# PIVOINE: Instruction Tuning for Open-world Entity Profiling

**Keming Lu**[†][*] **Xiaoman Pan**[‡]**, Kaiqiang Song**[‡]**, Hongming Zhang**[‡]
**Dong Yu**[‡]**, Jianshu Chen**[‡]

[†]University of Southern California, Los Angeles, CA
[‡]Tencent AI Lab, Bellevue, WA
[†]`keminglu@usc.edu`
[‡]`{xiaomanpan,riversong,hongmzhang}@global.tencent.com`
`{dyu,jianshuchen}@global.tencent.com`

## Abstract

This work considers the problem of Open-world Entity Profiling, which is a sub-domain of Open-world Information Extraction (Open-world IE). Unlike the conventional closed-world IE, Open-world IE considers a more general situation where entities and relations could be beyond a predefined ontology. We seek to develop a large language model (LLM) that can perform Open-world Entity Profiling with instruction tuning to extract desirable entity profiles characterized by (possibly fine-grained) natural language instructions. In particular, we construct INSTRUCTOPEN-WIKI, a substantial instruction-tuning dataset for Open-world Entity Profiling enriched with a comprehensive corpus, extensive annotations, and diverse instructions. We finetune pretrained BLOOM models on INSTRUCTOPEN-WIKI and obtain PIVOINE, an LLM for Open-world Entity Profiling with strong instruction-following capabilities. Our experiments demonstrate that PIVOINE significantly outperforms traditional methods and ChatGPT-based baselines, displaying impressive generalization capabilities on both unseen instructions and out-of-ontology cases. Consequently, PIVOINE emerges as a promising solution to tackle the open-world challenge in entity profiling. [1]

## 1 Introduction

Information extraction (IE) aims to discern meaningful information from unstructured data sources (Grishman, 2015). A traditional IE pipeline contains an array of tasks, which include, but are not limited to, Named Entity Recognition (NER) (Lample et al., 2016), Entity Linking (EL) (Kolitsas et al., 2018), Entity Typing (ET) (Ren et al., 2016), Relation Extraction (RE) (Huguet Cabot and Navigli, 2021), etc.

---

*Work done during Keming Lu's internship at Tencent AI Lab.

[1]Checkpoints and datasets are available at https://github.com/Lukeming-tsinghua/Instruction-Tuning-for-Open-world-IE

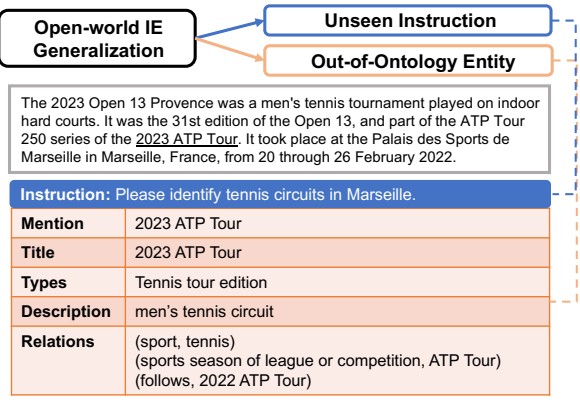

Figure 1: Illustration of Open-world Entity Profiling and its two main challenges: generalization to unseen instructions and out-of-ontology entities.

IE plays a vital role in knowledge graph construction (Schneider et al., 2022), search engine (Wang et al., 2022), and document analysis (Chiticariu et al., 2010; Wang et al., 2018; Zhong et al., 2020).

Most existing IE methods center around a **closed-world** setting with predefined ontologies. For instance, NER generally extracts named entities within several categories (Lample et al., 2016); EL focuses on associating mentions with a predefined ontology (Kolitsas et al., 2018). To better break this limitation, we introduce Open-world Entity Profiling, a series of entity-centric tasks within Open-world Information Extraction (Open-world IE) to accommodate broad and diverse requests related to entity profiles surpassing predefined ontologies' limits. Past research on open-setting IE has predominantly focused on individual tasks of IE, such as EL (Iurshina et al., 2022; Ruas and Couto, 2022) and OpenIE (Niklaus et al., 2018; Bhardwaj et al., 2019a). Consequently, a noticeable gap exists in comprehensive end-to-end studies aiming to create more extensive entity profiles within an open-world setting.

Furthermore, closed-world IE usually extracts all information without focusing on desired targets.

Therefore, we design Open-world Entity Profiling accepting an unstructured corpus and an instruction that characterizes target entities, identifies all entities within the context, and generates entity profiles, as shown in Fig. 1. With the emergence of large language models (LLMs) (Zhao et al., 2023), generative IE based on LLMs holds substantial promise in addressing the open-world challenge, given their exceptional generalization capabilities on various instructions. Open-world Entity Profiling can also be a pivotal capability for integrating plugins into the ChatGPT system since it provides a flexible communication interface between LLMs and their plugins. Nevertheless, existing research on LLMs reveals that they typically perform poorly in IE under zero-shot scenarios, necessitating appropriate instruction tuning to enhance their IE capabilities (Ma et al., 2023; Wadhwa et al., 2023). Therefore, instruction tuning (Wei et al., 2022) becomes critical in endowing LLMs with Open-world Entity Profiling abilities.

To generalize well on unseen instructions and ontologies, we develop PIVOINE (Instruction-following Open-world Information Extraction). PIVOINE is an LLM designed for Open-world Entity Profiling. We formulate Open-world Entity Profiling as an instruction-following auto-regressive generation task to generate comprehensive entity profiles in JSON. We explore eight popular categories of instructions in various granularities. Each category imposes specific constraints on candidate entities. In pursuit of generalization over unseen cases, we develop an instruction tuning dataset INSTRUCTOPENWIKI for Open-world Entity Profiling, including diverse instructions that endow PIVOINE with solid instruction-following capability. INSTRUCTOPENWIKI incorporates rich annotations, various instructions, and a delicate out-of-ontology evaluation set, significantly contributing to the generalization of unseen cases.

The contributions of this work are three-fold. First, we define Open-world Entity Profiling and develop PIVOINE, which performs entity-centric IE without the limitations of predefined ontology. This flexibility allows for its generalization abilities and application across diverse downstream scenarios. Second, we construct a substantial Open-world Entity Profiling dataset INSTRUCTOPENWIKI. Third, we explore a comprehensive evaluation for Open-world Entity Profiling. We meticulously design an open-world evaluation set incorporated in IN-STRUCTOPENWIKI to assess Open-world capabilities thoroughly, focusing on the generalization of unseen instructions and out-of-ontology entities. Our contributions are verified with experiments and multifaceted analysis. Most notably, PIVOINE exhibits impressive generalization capabilities on unseen instructions and out-of-ontology cases, demonstrating its robust potential to address the open-world challenge effectively.

## 2 Related Works

**Large Language Models.** Large language models (LLMs) is an emerging topic summarized in a recent survey Zhao et al. (2023). Therefore, we only provide a highly-selective review. Brown et al. (2020) train an auto-regressive language model GPT-3 with 175 billion parameters, showing extraordinary task-agnostic few-shot performance. Chowdhery et al. (2022) develop a Pathways Language Model PALM and scale it up to 540 billion parameters. Scao et al. (2022) propose BLOOM, open-access LLMs from 560 million to 175 billion parameters. Touvron et al. (2023a) develop LLAMA, a more efficient public-accessible LLM. We use BLOOM as the backbone since it was the latest public LLM pretrained on a diverse corpus, including codes, when we conducted this study. However, other latest LLMs, such as LLAMA, can also be easily tuned on our dataset to acquire open-world IE abilities.

**Instruction Tuning.** Instruction tuning is an emergent paradigm that finetunes LLMs on datasets described by instructions. Wei et al. (2022) finetune an LLM with 175 billion parameters on various NLP datasets with instruction templates and proof instruction tuning can significantly improve zero-shot performance. Ouyang et al. (2022) show supervised instruction tuning and finetuning with human feedback helps LLMs align with human intent. This work is further extended by OPENAI and becomes the product CHATGPT[2] used as a baseline in our work. In this work, we create an instruction-following dataset INSTRUCTOPENWIKI for open-world IE and employ instruction tuning to empower LLMs with Open-world Entity Profiling abilities.

**Information Extraction.** Instruction-following IE reformulates IE into a generation task with instructions describing target information. We mainly present two concurrent works as this is an emerg-

---

[2] https://openai.com/blog/chatgpt

ing topic. Wei et al. (2023) solve IE as a multi-turn question-answering format by providing pre-defined instructions to CHATGPT. Wang et al. (2023) proposes an instruction-tuning IE benchmark and develops a unified IE method. However, all these works are based on the closed-world setting and have not adapted to Open-world IE, precisely our focus in this work. To our best knowledge, PIVOINE is the first work exploring instruction-following open-world entity profiling. Previous explorations are limited to sub-fields of IE, such as the NIL problem in EL (Lin et al., 2012) and open information extraction (Zhou et al., 2022).

## 3 Methods

We describe PIVOINE, an Open-world Entity Profiling method with instruction-following abilities. We will introduce preliminaries (§3.1), instruction-following open-world entity profiling (§3.2), and construction of the dataset INSTRUCTOPENWIKI (§3.3).

### 3.1 Preliminaries

**Problem Definition.** Open-world Entity Profiling aims to extract entity profiles from unstructured texts without predefined ontologies by following specific instructions. Entity profiles include mentions, canonical titles, types, descriptions, aliases, and relations, as shown in Fig. 1. Specifically, mentions are text spans linked to entities; types are a list of phrases an entity is an instance of; aliases are a list of synonyms; relations are a list of relation titles between extracted entities within the input. Given a document and an instruction describing a specific constraint about target entities, Open-world Entity Profiling methods are expected to generate entity profiles that fix the instruction.

**Method Overview.** We solve Open-world Entity Profiling by instruction tuning of LLMs. As shown in Fig. 2, we first reformulate Open-world Entity Profiling into auto-regressive generation by linearizing the structure knowledge into the JSON format (§3.2). We apply instruction tuning to empower PIVOINE to extract different entities following instructions in different granularities. To do so, we build INSTRUCTOPENWIKI, a large-scale instruction-following Open-world Entity Profiling dataset. As presented in Fig. 3, INSTRUCTOPENWIKI is created by weak supervision between large-scale corpus and existing knowledge base (§3.3).

Then, we augment the dataset with diverse instructions and rephrase them to enrich semantic diversity. We also comprehensively evaluate PIVOINE on the open-world evaluation set (§4).

### 3.2 Instruction Tuning for Open-world Entity Profiling

Leveraging the strong generalization abilities of LLMs to pursue generalization on unseen instructions and out-of-ontology cases, we reformulate Open-world Entity Profiling as an instruction-following generation task. We create diverse instructions and linearize structured IE outputs into JSON sequences. And then, we finetune LLMs in a supervised instruction tuning (SFT) setup, training LLMs to generate the targeted JSON sequence of entity profiles as the output.

**Instruction Design.** The diversity of instruction categories is essential for the generalization of unseen instructions. We manually designed eight instruction categories with varying granularities:

- **Default**: Extract all entities and relations in the input without additional requirements.
- **Base Type**: Extract entities of given base types. We define base types as fine-grained types in Wikidata, building from the "P31 (instance of)" properties of entities in Wikidata.
- **Abstract Type**: Extract entities of given abstract types. We define abstract types as more coarse-grained types obtained by finding parent "P279 (subclass of)" properties of base types in Wikidata. This instruction category is only designed for extending type semantics during the training and helps LLMs learn type hierarchy. Both instructions with types share the same prompt, so we do not distinguish them in inference.
- **Description**: Extract entities that fit given descriptions, providing ultra fine-grained instructions that require Open-world Entity Profiling methods directly understand diverse descriptions. As shown in Fig. 2, descriptions can be phrases or sentences describing entities' properties.
- **Importance**: Extract the top-K most important entities. The importance of entities is defined as entity priorities in Wikidata. This instruction requires Open-world Entity Profiling methods to rank entities with inherent priorities properly.
- **Number**: Extract a specific number of entities in the input document. Unlike other categories, instructions with number aim to extract partial information from the input, and the answer is not

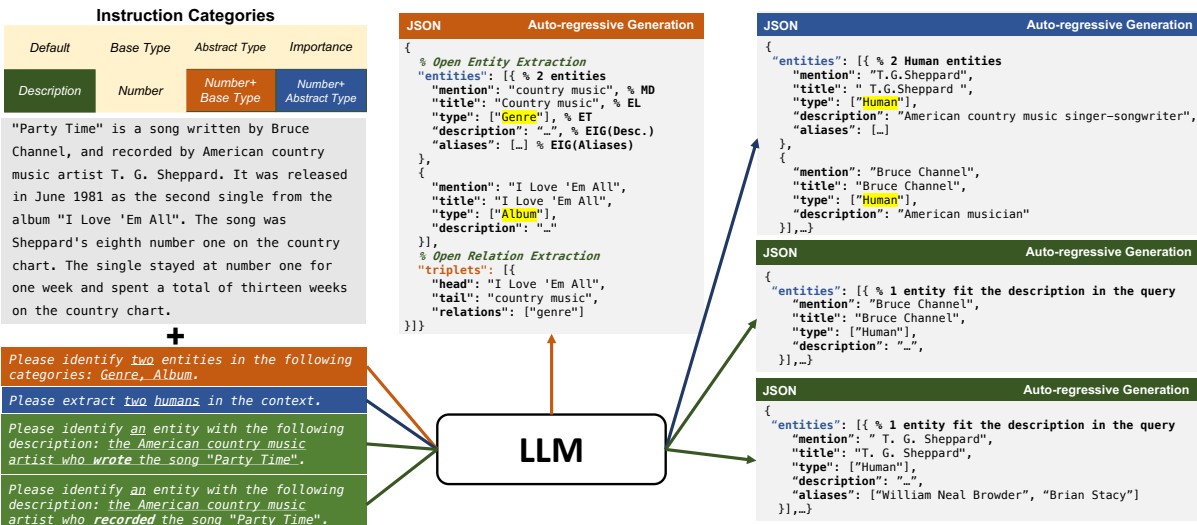

Figure 2: Overview of the open-world entity profiling method PIVOINE. This figure shows four generation cases of PIVOINE under three instruction categories colored in orange, blue, and green. PIVOINE takes corpus and instructions as inputs and auto-regressively generates JSON sequences. PIVOINE can extract different entity profiles based on different instructions from the same corpus. The auto-regressive generation of JSON targets aligns well with various IE tasks marked as green comments in the JSON.

unique. Therefore, we separately analyze these categories in evaluation.

- **Number+Base Type**: Cross instructions between categories **Number** and **Base Type**.
- **Number+Abstract Type**: Cross instructions between categories **Number** and **Abstract Type**.

We manually design an instruction template for each category. Then we ask CHATGPT to rephrase the manually designed templates, enhancing the semantic diversity of these templates. Details of original and rephrased templates are shown in Tab. 8. CHATGPT prompts for rephrasing are shown in Tab. 22. We train PIVOINE to follow single instructions. And we add the last two cross-instructions to evaluate generalization on complex instructions.

**Linearization.** Previously, various techniques have been explored to linearize the structured information in the generative IE (Ye et al., 2022) but either lack semantics or require additional training for special tokens (Lou et al., 2023; Wang et al., 2021; Lu et al., 2022). To better leverage pretrained knowledge, we present a novel idea that uses the JSON (JavaScript Object Notation) format to linearize heterogeneous structured entity profiles. JSON is primarily used to transmit data in web applications, so it frequently appears in codes. Therefore, LLMs pretrained on codes are familiar with the JSON schema, avoiding additional training for special tokens or manually-defined structure templates. Furthermore, JSON uses a text-based syntax with

key-value pairs, capturing additional semantics in natural language by keys and providing flexible structures. This linearization aggregates multiple IE subtasks, revealing the chain of thoughts in IE employed in traditional pipeline methods.

### 3.3 Instruction Dataset Construction

Learning from a large-scale instruction tuning dataset with a rich corpus and large ontology contributes to the generalization of out-of-ontology cases. However, building a large-scale Open-world Entity Profiling dataset by manual annotations is infeasible since identifying entities in text and linking them with entity profiles require tremendous human effort. Therefore, we develop a weakly supervised method that automatically creates the dataset INSTRUCTOPENWIKI for instruction tuning.

**Weak Supervision.** INSTRUCTOPENWIKI is created by aligning anchor links in Wikipedia[3] to entity profiles in its corresponding knowledge base Wikidata[4] by the wiki identifiers, shown in the left part of Fig. 3. Wikipedia is a large corpus covering various domains, while Wikidata contains rich world knowledge. Wikipedia and Wikidata are frequently revised by contributors worldwide, ensuring precision and being up-to-date. All anchor links in Wikipedia are manually annotated, so link-

---

[3] https://www.wikipedia.org/
[4] https://www.wikidata.org

ing between mentions and entities is reliable. We only use the leading paragraph in each Wikipedia article since it contains the richest anchor links. Besides, anchor links referring to the same entity may only be marked once the first time within an article, so using the rest of the paragraphs will face higher risks of missing mention annotations. We retrieve four fields from Wikidata as its profile for each entity, where canonical titles are English labels of entities, and types are derived from "instance of (P31)" properties. After identifying all entities in a paragraph, we employ distant supervision to identify relations between these entities from Wikidata as described at the top of Fig. 3. Specifically, we link a relation triplet in the KB to this paragraph if both head and tail entities are mentioned. A relation triplet is represented by mentions of head and tail entities and a list of relation names. Detailed statistics are presented in Appx. §B.

**Instruction Augmentation.** We further augment the dataset with predefined instructions as shown in the middle of Fig. 3. We generate an instruction-tuning sample with the default instruction and randomly augment one another from six training categories. All instructions focus on entities, and we also filter triplets to ensure alignment.

## 4   Evaluation

A well-designed open-world evaluation set is essential for evaluating Open-world Entity Profiling methods without bias. So we create an open-world evaluation set with rich out-of-ontology cases and design metrics for evaluating such performance.

**Open-world Evaluation Set.** Previous work constructs open-world test sets by simply holding out a portion of entities from the training ontology. However, such methods may introduce potential risks and lead to insufficient evaluation. First, holding out entities from the training ontology also removes corresponding links in the training corpus, hindering the completeness of annotations. Moreover, mentions of held-out entities still frequently appear in the training corpus even if they are not explicitly annotated, resulting in potential data leakage. Therefore, we propose a delicate method that uses the time difference between Wiki dumps to construct a strictly open-world test set. As shown in the bottom of Fig. 3, we use the Wikidata dump on 05/30/2022 and the Wikipedia dump on 06/20/2022 to build the training set. As the evaluation corpus, we filter all new articles between two Wikipedia

dumps, 06/20/2022 and 03/01/2023. And we select new entities appearing in the Wikidata dump on 03/01/2023 but not presented in the Wikidata on 05/30/2022 as out-of-ontology entities [5]. Using time difference to build an out-of-ontology evaluation set minimizes potential data leakage and maximizes completeness of training annotations.

**Metrics.** Although defining Open-world Entity Profiling as an end-to-end task, we still split it into six tasks in evaluation to provide more comprehensive analyses: **(1) Mention Detection (MD)** corresponds to the correctness of mention extraction in "mention" keys. **(2) Entity Linking (EL)** evaluating whether models generate proper canonical titles for mentions in "title" keys. We use hard and soft matching based on a ROUGE-L F1 threshold as the criterion. **(3) Entity Typing (ET)** requires models generate entity types. **(4) Open Relation Extraction (RE)** evaluates extracted relations under "triplets". Inspired by OpenIE (Zhou et al., 2022), we use CaRB with the ROUGE-L matcher (Bhardwaj et al., 2019b) to evaluate triplet generation performance. **(5) Description Generation (EIG-Desc.)** requires models to generate description. **(6) Aliases Generation (EIG-Aliases)** expects models to generate aliases in "aliases" keys. We report precision, recall, and F1 scores on each task except description generation, where we report ROUGE-L instead. We randomly select three rephrased templates for each sample during evaluation and report the average with standard deviation.

**Unseen Ontologies.** We explore the out-of-ontology generalization by separately analyzing the recall of training (Before 05/30/2022) and out-of-training (After 05/30/2022) entities. For instance, *2023 ATP Tour (Q111441127)* shown in Fig. 1 is an unseen entity introduced to Wikidata after 05/20/2022. Open-world Entity Profiling methods are proven to have a great generalization of unseen ontologies if they can extract this entity from the latest corpus.

**Unseen Instructions.** We also split the test set into samples with unseen and seen instructions under the most fine-grained category "Description". Unseen instructions query target entities that are not in the training instructions. The proportions of queries with unseen entities are shown in Tab. 4.

---

[5]The ROOTS corpus (Laurençon et al., 2022), pretrained corpus of BLOOM, only includes Wikipedia dump before 06/20/2022, so this evaluation set also remains unseen for the BLOOM pretraining.

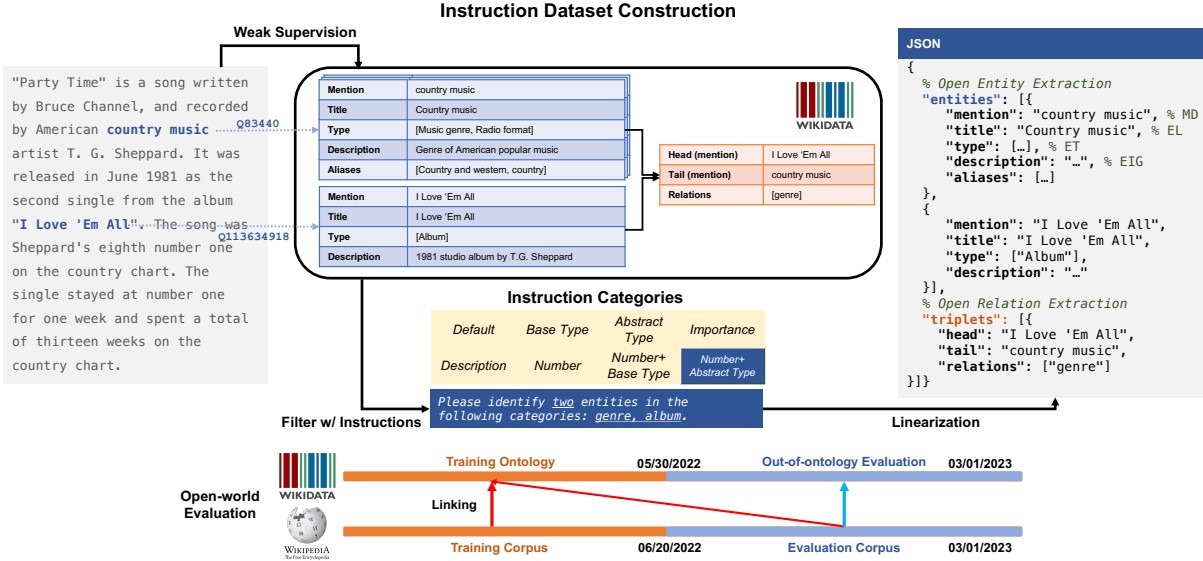

Figure 3: Overview of instruction-following open-world Entity Profiling dataset construction. The top of this figure shows INSTRUCTOPENWIKI is created by aligning anchor links in Wikipedia to entities and relations in the knowledge base Wikidata, then augmented with instructions within eight categories. Entity profiles are identified from Wikidata and linearized into a sequence in the JSON format. The bottom of this figure shows how we create an open-world evaluation set with the time difference between Wiki dumps.

Furthermore, we create three more instruction templates by manually writing and GPT-4 generation, respectively, and make sure they have no overlap with rephrased training instructions. So we can analyze the performance difference between the two new sources and training instructions. We also separately evaluate number-related instructions as these partial extraction instructions have no unique correct answers.

## 5 Experiments

We carry out a comprehensive evaluation of PIVOINE, including experimental setup (§5.1), main results (§5.2), and further analysis (§5.3).

### 5.1 Experimental Setup

**Baselines.** We employ ChatGPT as our main baseline since no previous Open-world Entity Profiling methods exist to our best knowledge. **(1) CHAT-GPT** is an instruction-following LLM that can handle various tasks. Detailed configurations of CHATGPT are described in Appx. §C. **(2) CHAT-GPT W/DEMO** is a stronger baseline with a one-shot demo based on CHATGPT. We provide CHAT-GPT an instruction example in the conversation history. Furthermore, we also involve the latest VI-CUNA-7B-v1.5 (Zheng et al., 2023) as a baseline, an open-source instruction-tuning model trained on over 150k dialogues based on the foundation model LLaMa-2 (Touvron et al., 2023b). We also introduce two traditional closed-world IE baselines: **(3) GENRE** (Kumar and Bojar, 2022) is the first system retrieving entities by generating canonical names, which can address MD and EL but is constrained by the KILT entities. **(4) OPENIE6** (Kolluru et al., 2020) is a well-developed neural OpenIE system. We employ it as a baseline of RE.

**Configurations.** We start from BLOOM (Scao et al., 2022) checkpoints with 1 billion and 7 billion parameters and run instruction tuning on INSTRUCTOPENWIKI. We use suggested hyperparameters for finetuning each model in Scao et al. (2022). PIVOINE-1b is trained on 64 NVIDIA V100 GPU for 92 hours. PIVOINE-7b is trained on 256 NVIDIA V100 GPU for 54 hours. Detailed configurations are shown in Appx. §E.

### 5.2 Main Results

We present our main results in three aspects: overall performance, generalization study on unseen ontologies, and unseen instructions.

**Overall Performance.** Tab. 1 shows overall performance on six tasks. We report macro average F1 scores with all instruction categories. And we only report performance on default instruction for GENRE and OPENIE6 for reference since they have no instruction-following abilities. They fail to achieve outstanding performance on correspond-

| Method | MD F1 | EL F1(T=1) | EL F1(T=0.8) | ET F1 | OpenRE F1(CaRB) | EIG (Desc.) F1(ROUGE-L) | EIG (Aliases) F1 |
|---|---|---|---|---|---|---|---|
| Closed-world Baselines | $43.7^\dagger$ | $17.2^\dagger$ | $20.1^\dagger$ | – | $15.2^\dagger$ | – | – |
| CHATGPT | $50.5_{0.2}$ | $25.0_{0.2}$ | $26.1_{0.2}$ | $7.9_{0.0}$ | $22.5_{0.2}$ | $39.0_{0.2}$ | $15.8_{0.0}$ |
| CHATGPT W/DEMO | $51.1_{0.1}$ | $38.6_{0.0}$ | $39.9_{0.0}$ | $14.8_{0.1}$ | $21.4_{0.0}$ | $52.0_{0.2}$ | $13.0_{0.1}$ |
| VICUNA-7B | $30.6_{0.0}$ | $16.5_{0.0}$ | $17.2_{0.0}$ | $3.4_{0.0}$ | $20.0_{0.0}$ | $28.4_{0.0}$ | $10.8_{0.1}$ |
| PIVOINE-1b | $61.4_{0.0}$ | $49.6_{0.0}$ | $50.6_{0.0}$ | $40.4_{0.0}$ | $56.1_{0.2}$ | $68.6_{0.0}$ | $72.7_{0.1}$ |
| PIVOINE-7b | $\mathbf{79.6_{0.0}}$ | $\mathbf{69.8_{0.1}}$ | $\mathbf{70.7_{0.1}}$ | $\mathbf{56.4_{0.0}}$ | $\mathbf{67.8_{0.2}}$ | $\mathbf{80.2_{0.1}}$ | $\mathbf{80.5_{0.0}}$ |

Table 1: Main results of overall performance in end-to-end evaluation. We report the macro average of F1 scores with all instruction categories on mention detection (MD), entity linking (EL), entity typing (ET), open relation extraction (OpenRE), and entity information generation (EIG) for descriptions and aliases. Subscript scores are the deviation under three different rephrased instructions for each sample. Comprehensive results are demonstrated in Appx. §F. † We only report the performance of closed-world baselines on default instructions since they focus on specific tasks without instruction following abilities, such as GENRE in MD, EL, ET and OPENIE in OpenRE.

| Partition | Method | MD R | EL R(T=1) | EL R(T=0.8) | ET R | OpenRE R(CaRB) | EIG (Desc.) R(ROUGE-L) | EIG (Aliases) R |
|---|---|---|---|---|---|---|---|---|
| | Closed-world Baselines | $55.3^\dagger$ | $23.8^\dagger$ | $24.7^\dagger$ | – | $21.9^\dagger$ | – | – |
| | CHATGPT | $59.3_{0.3}$ | $31.8_{0.2}$ | $32.3_{0.2}$ | $9.7_{0.1}$ | $34.1_{0.2}$ | $40.2_{0.4}$ | $10.1_{0.1}$ |
| Before | CHATGPT W/DEMO | $56.5_{0.0}$ | $45.0_{0.1}$ | $45.7_{0.1}$ | $15.9_{0.2}$ | $38.3_{0.0}$ | $53.6_{0.2}$ | $8.4_{0.0}$ |
| 05/30/2022 | VICUNA-7B | $26.5_{0.0}$ | $14.9_{0.0}$ | $15.1_{0.0}$ | $3.3_{0.0}$ | $36.4_{0.0}$ | $29.4_{0.0}$ | $8.8_{0.0}$ |
| | PIVOINE-1b | $59.2_{0.1}$ | $51.8_{0.0}$ | $52.2_{0.0}$ | $44.4_{0.1}$ | $61.8_{0.3}$ | $71.4_{0.1}$ | $69.7_{0.1}$ |
| | PIVOINE-7b | $\mathbf{83.7_{0.0}}$ | $\mathbf{78.8_{0.1}}$ | $\mathbf{79.1_{0.1}}$ | $\mathbf{66.8_{0.1}}$ | $\mathbf{79.4_{0.1}}$ | $\mathbf{82.5_{0.0}}$ | $\mathbf{78.0_{0.0}}$ |
| | Closed-world Baselines | $33.6^\dagger$ | $0^\dagger$ | $3.6^\dagger$ | – | $18.2^\dagger$ | – | – |
| | CHATGPT | $58.9_{0.2}$ | $23.8_{0.2}$ | $26.5_{0.3}$ | $6.9_{0.1}$ | $27.1_{0.5}$ | $36.0_{0.3}$ | $14.3_{0.3}$ |
| After | CHATGPT W/DEMO | $58.2_{0.0}$ | $39.1_{0.0}$ | $42.1_{0.0}$ | $14.2_{0.1}$ | $29.3_{0.5}$ | $48.1_{0.2}$ | $13.3_{0.1}$ |
| 05/30/2022 | VICUNA-7B | $26.7_{0.0}$ | $13.0_{0.0}$ | $14.3_{0.0}$ | $3.1_{0.0}$ | $25.5_{0.0}$ | $25.8_{0.0}$ | $10.5_{0.0}$ |
| | PIVOINE-1b | $55.1_{0.0}$ | $36.3_{0.1}$ | $38.3_{0.1}$ | $22.8_{0.0}$ | $34.5_{0.1}$ | $55.7_{0.3}$ | $20.5_{0.2}$ |
| | PIVOINE-7b | $\mathbf{71.7_{0.1}}$ | $\mathbf{51.7_{0.2}}$ | $\mathbf{53.8_{0.2}}$ | $\mathbf{24.1_{0.2}}$ | $\mathbf{36.1_{0.1}}$ | $\mathbf{64.1_{0.2}}$ | $\mathbf{22.7_{0.1}}$ |

Table 2: Main results of generalization analysis in end-to-end evaluation. Headers are the same as Tab. 1 except we report recalls in each task. The partition "Before 05/30/2022" denotes mentions linked to the training ontology (seen entities), while "After 05/30/2022" denotes mentions linked to entities that are out of training ontology (unseen entities). Comprehensive results are demonstrated in Appx. §F. † We only report the performance of closed-world baselines on default instructions since they focus on specific tasks without instruction following abilities, such as GENRE in MD, EL, ET and OPENIE in OpenRE.

ing tasks, showing Open-world Entity Profiling is challenging for closed-world IE methods as we expect. CHATGPT can partially address Open-world Entity Profiling but only has 7.9% F1 score in ET and 15.8% F1 score in EIG(Aliases). CHATGPT W/DEMO significantly outperforms CHATGPT in EL and EIG(Desc.) and has comparable performance on other tasks, showing the demo in history benefits CHATGPT in Open-world Entity Profiling. PIVOINE-1b outperforms CHATGPT W/DEMO by about 10% absolute improvement in F1 over all tasks. PIVOINE-7b achieves the best performance among all methods, significantly outperforming PIVOINE-1b by nearly 20% F1 in most tasks. This result suggests larger models will boost the performance and even can potentially address some of the tasks.

**Generalization to unseen ontologies.** We separately analyze recalls of entities before and after 05/30/2022 in the open-world test set and present results in Tab. 2 to evaluate generalization on unseen ontologies. The partition "Before 05/30/2022" denotes mentions linked to the training ontology while "After 05/30/2022" denotes those that are out of training ontology. We first witness a consistent performance drop on unseen entities for all methods, especially closed-world baselines. Even for two ChatGPT-based baselines, performance on EL, ET, OpenRE, and EIG(Desc.) also drop dramatically. PIVOINE-1b outperforms CHATGPT W/DEMO on four tasks in the unseen partition, and surpasses CHATGPT W/DEMO on all tasks in the seen partition. PIVOINE-7b still achieves the best performance on tasks in both partitions, showing it can successfully generalize to out-of-ontology entities in Open-world Entity Profiling.

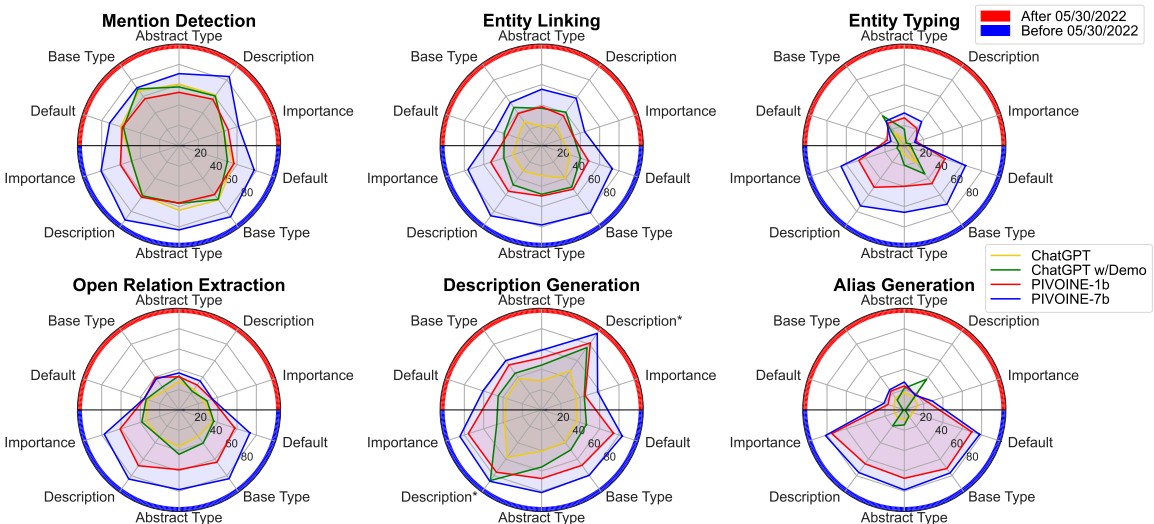

Figure 4: Main results of the **ontology generalization** for each instruction category in each task. The partition "Before 05/30/2022" denotes mentions linked to the training ontology (seen entities), while "After 05/30/2022" denotes mentions linked to entities that are out of training ontology (unseen entities). [*] Descriptions as instructions are not considered in description generation tasks to avoid data leakage. Original scores are shown in Appx. §F.

| Instruction Source | Partition | MD F1 | EL F1(T=1) | EL F1(T=0.8) | ET F1 | OpenRE F1(CaRB) | EIG (Aliases) F1 |
|---|---|---|---|---|---|---|---|
| Training | | $83.6_{0.2}$ | $55.1_{0.1}$ | $57.8_{0.1}$ | $28.2_{0.1}$ | $35.0_{0.3}$ | $12.8_{0.3}$ |
| Manual Written | $\Delta$ | $83.3_{0.2}$ $-0.3$ | $54.4_{0.2}$ $-0.7$ | $57.1_{0.1}$ $-0.7$ | $28.4_{0.1}$ $-0.2$ | $34.5_{0.1}$ $-0.5$ | $13.0_{0.2}$ $+0.2$ |
| GPT-4 Generation | $\Delta$ | $83.4_{0.2}$ $-0.2$ | $54.8_{0.1}$ $-0.3$ | $56.3_{0.1}$ $-0.4$ | $27.8_{0.1}$ $-0.4$ | $34.6_{0.2}$ $-0.4$ | $12.7_{0.2}$ $-0.1$ |

Table 3: Main results of the **instruction generalization** on "Description" instruction category with PIVOINE-7b on the unseen entities described in §4. $\Delta$ refers to absolute differences between instructions from new sources and training instructions. Headers are the same as Tab. 1. To avoid data leakage, descriptions as instructions are not considered in description generation tasks. Subscript scores are the deviation under three different rephrased instructions for each sample. The best scores in each partition are marked in **bold**.

Fig. 4 shows a more detailed analysis of each instruction category. Comparing six radar charts, all methods are easy to generalize to out-of-ontology entities on MD, EL, and EIG(Desc.) as their upper and lower parts of plots are nearly symmetric. However, such generalization is harder in RE since it is an end-to-end task requiring precise out-of-ontology MD first. ET and EIG(Aliases) are the most challenging for all methods because they require a comprehensive list of types and aliases. We also find PIVOINE-7b consistently outperforms other baselines on all instruction categories.

**Generalization to unseen instructions.** We evaluate the generalization on unseen instructions and present the results in Tab. 3. As introduced in the last paragraph of §4, unseen instructions are constructed by manual writing and GPT-4. We eval-

uate the performance difference of PIVOINE-7b with unseen entities queried by instructions from various sources of instructions. The performance difference on most tasks is within 1 absolute percent of F1 scores. Therefore, PIVOINE-7b shows extraordinary generalization to unseen instructions in Open-world Entity Profiling.

### 5.3 Analysis

We provide further analyses to reveal model behaviors on Open-world Entity Profiling.

**Instruction Following.** We analyze instruction following qualities in three aspects as shown in Fig. 5. First, we analyze the JSON decoding error rates, as generating in the correct JSON schema is essential for parsing extracted information. As the left figure in Fig. 5, ChatGPT-based baselines fail to ensure

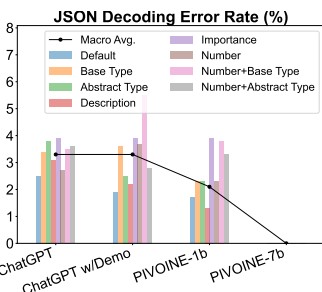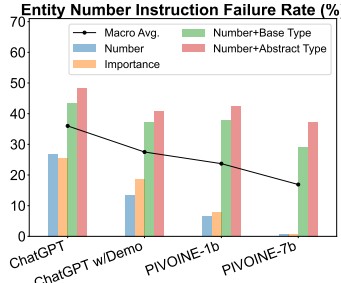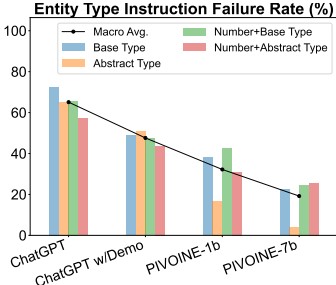

Figure 5: Analysis of the instruction-following capabilities, including JSON decoding error rate, entity number instruction failure rate, and entity type instruction failure rate. Original scores are shown in Appx. §F.

the valid JSON schema, especially on unseen cross instructions. PIVOINE-1b is better than baselines on average but still faces decoding errors. However, PIVOINE-7b is a trustworthy model with no JSON decoding errors on all instruction categories, even for unseen cross instructions.

We also compare entity number instruction failure rates on four number-related instruction categories as shown in the middle of Fig. 5. PIVOINE-7b still has the lowest failure rates in extracting the correct number of entities, which is close to zero on two trained instruction categories. All methods fail to follow number instructions when receiving untrained cross instructions, suggesting cross instructions are still challenging for current methods. Furthermore, results in Tab. 21 shows that number-related instructions do not hinder the precision of MD and EL. Instructions with number constraints provide partial extraction abilities to PIVOINE so that we can control the precision-recall trade-off by specifying the number in instructions.

Similar patterns are observed in entity type instruction failure rate analysis in the right part of Fig. 5. PIVOINE-7b only has half failure rates on cross instruction categories compared with the vanilla CHATGPT, showing PIVOINE-7b has much better type instruction following abilities. We also notice that following instructions with abstract types is significantly easier for PIVOINE than fine-grained base types.

**Human Evaluation.** PIVOINE may extract correct relation triplets that are out of the scope of existing Wikidata. Therefore, we randomly select 100 relation triplets with unseen entities predicted by PIVOINE-7b to analyze its precision further. We consider the evidence of a relation triplet as the context from which PIVOINE-7b extracts the triplet. To minimize the manual effort, we first reformulate the output of PIVOINE-7b to a prompt and ask GPT-

4 to provide judgment and explanation based on evidence supplied by PIVOINE. We also manually evaluate the correctness of relation triplets based on the same evidence without additional world knowledge. The accuracy of relation triplets provided by GPT-4 is 87%, and manual checking accuracy is 83%. The agreement between GPT-4 and the annotator is 87%, suggesting GPT-4 is capable of evaluating relation correctness. The latest Wikidata also verifies 8% cases. This evaluation shows PIVOINE-7b can precisely excavate new entities and relations from the corpus.

## 6 Conclusion

We propose Open-world Entity Profiling, a challenging task that aims to extract out-of-ontology entity profiles with instructions. Towards this grand mission, we create a large-scale Open-world Entity Profiling dataset INSTRUCTOPENWIKI and develop PIVOINE by instruction tuning. We conduct extensive experiments on diverse instruction categories and different model scales, showing PIVOINE is a trustworthy LLM capable of following (possibly unseen) instructions in various granularities and extracting out-of-ontology entity profiles. Valuable future works include extending PIVOINE to a larger scale and exploring more comprehensive instruction categories.

## Limitations

The contents in PIVOINE outputs are limited by the output length constraint of BLOOM. However, techniques for extending such length limitation are rapidly developed (Yu et al., 2023). Furthermore, instructions investigated in this work mainly focus on entities. Future works can extend the scope of instruction categories to cover demands about relations or events.

## Ethics statement

All data and results are obtained and processed according to the respective data and API usage policy. PIVOINE is developed based on BLOOM (Scao et al., 2022). Although PIVOINE has great performance in Open-world Entity Profiling, PIVOINE may still generate potential counterfactual outputs. Outputs from PIVOINE need careful verification before being applied to real-world applications.

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

# Appendix

## A  Detailed Instructions

Tab. 8 shows original manually-designed seed instruction, rephrased instructions, and number of rephrased instructions for each category. We use ChatGPT to rephrased our original instructions. The prompt we used is shown in Tab. 22.

## B  Dataset Statistics

We display the statistics of INSTRUCTOPENWIKI in Tab. 9. INSTRUCTOPENWIKI contains a rich corpus, which includes all head paragraphs of all Wikipedia articles with 39 million mentions and 19 million triplets. And INSTRUCTOPENWIKI is also annotated with a large ontology containing over 2 million entities with 21 thousand entity types and 962 relation types, ensuring it covers a wide range of domains. Besides, the entity information density of INSTRUCTOPENWIKI is also abundant, so models can be trained for extracting entity profiles efficiently.

| Partition | Base Type | Abstract Type | Description |
|---|---|---|---|
| Unseen | 18.9% | 17.4% | 82.3% |
| Seen | 81.1% | 82.6% | 17.7% |

Table 4: Proportions of unseen and seen instructions in "Base Type", "Abstract Type", and "Description" in the test set.

We also create a small development set to select the best checkpoint balancing performance on seen and unseen cases. As shown in Tab. 9, the open-world evaluation test set contains rich unseen mentions (27.1%, 29,612/109,411) annotated by unseen entities (24.2%, 9,464/39,086).

## C  CHATGPT Configurations

We provide ChatGPT with the input context, instruction, and output JSON schema to prompt ChatGPT to solve instruction-following open-world IE. We use ChatGPT in May 2022 and query it with the official API provided by OpenAI[6]. The detailed prompt we used is shown in Tab. 23.

## D  Details of Human Evaluation

We use the May 1st, 2023 version of GPT-4. We query it via the chat platform of OpenAI[7].

[6]https://openai.com/blog/introducing-chatgpt-and-whisper-apis
[7]https://chat.openai.com/

## E  Hyper-parameters

We develop our models with Megatron-Deepspeed from BigScience. We trained PIVOINE-1b for 10,294 steps with a global batch size of 1,024 and PIVOINE-7b for 5,000 steps with a global batch size of 2,048. The training steps and learning rates are selected by performance on the development set with grid search. We infer on 256 NVIDIA V100 GPU within 30 minutes with the official BLOOM inference project provided by HuggingFace[8]. We generate with beam search without sampling and the number of beams is 4. The maximum number of generated tokens is 2,048. The other generation parameters are set to default.

## F  Comprehensive Results

| Instruction | ChatGPT | ChatGPT w/Demo | PIVOINE-1b | PIVOINE-7b |
|---|---|---|---|---|
| Default | $2.5_{0.1}$ | $1.9_{0.1}$ | $1.7_{0.0}$ | $0.0_{0.0}$ |
| Base Type | $3.4_{0.3}$ | $3.6_{0.4}$ | $2.3_{0.2}$ | $0.0_{0.0}$ |
| Abstract Type | $3.8_{0.4}$ | $2.5_{0.8}$ | $2.3_{0.2}$ | $0.0_{0.0}$ |
| Description | $3.1_{0.3}$ | $2.2_{0.2}$ | $1.3_{0.3}$ | $0.0_{0.0}$ |
| Importance | $3.9_{0.6}$ | $3.9_{0.9}$ | $3.9_{0.3}$ | $0.0_{0.0}$ |
| Number | $2.7_{0.5}$ | $3.7_{0.1}$ | $2.3_{0.1}$ | $0.0_{0.0}$ |
| Number+ Base Type | $3.5_{0.2}$ | $5.5_{0.3}$ | $3.8_{0.1}$ | $0.0_{0.0}$ |
| Number+ Abstract Type | $3.6_{0.4}$ | $2.8_{1.0}$ | $3.3_{0.4}$ | $0.0_{0.1}$ |
| Macro Avg. | $3.3_{0.2}$ | $3.3_{0.4}$ | $2.1_{0.0}$ | $0.0_{0.0}$ |

Table 5: Analysis of JSON format correctness on all instruction categories. We report the JSON decoding error rates (%) in this table. Subscript scores are the deviation under three different rephrased instructions for each sample.

| Instruction | ChatGPT | ChatGPT w/Demo | PIVOINE-1b | PIVOINE-7b |
|---|---|---|---|---|
| Number | $26.8_{0.4}$ | $13.4_{0.5}$ | $6.5_{0.0}$ | $0.6_{0.0}$ |
| Importance | $25.6_{1.1}$ | $18.7_{0.2}$ | $7.9_{0.3}$ | $0.8_{0.0}$ |
| Number+ Base Type | $43.4_{0.4}$ | $37.3_{0.0}$ | $38.0_{0.9}$ | $29.0_{0.2}$ |
| Number+ Abstract Type | $48.3_{0.9}$ | $40.8_{0.4}$ | $42.3_{0.8}$ | $37.3_{0.1}$ |
| Macro Avg. | $36.0_{0.1}$ | $27.5_{0.0}$ | $23.7_{0.1}$ | $16.9_{0.0}$ |

Table 6: Analysis of following entity number constraints. We report error rates (%) of predictions that do not have the same number of entities as the instruction. Subscript scores are the deviation under three different rephrased instructions for each sample.

[8]https://github.com/huggingface/transformers-bloom-inference

| Instruction | ChatGPT | ChatGPT w/Demo | PIVOINE-1b | PIVOINE-7b |
|---|---|---|---|---|
| Base Type | $72.4_{0.8}$ | $48.8_{0.1}$ | $38.1_{0.3}$ | $22.4_{0.6}$ |
| Abstract Type | $65.1_{2.1}$ | $50.7_{0.7}$ | $16.9_{0.3}$ | $4.1_{0.2}$ |
| Number+ Base Type | $65.8_{0.5}$ | $47.3_{0.4}$ | $42.5_{0.8}$ | $24.7_{0.3}$ |
| Number+ Abstract Type | $57.1_{1.6}$ | $43.5_{0.8}$ | $31.1_{0.7}$ | $25.5_{1.5}$ |
| Macro Avg. | $65.1_{0.8}$ | $47.6_{0.1}$ | $32.2_{0.5}$ | $19.2_{0.5}$ |

Table 7: Analysis of following entity type constraints. We report error rates (%) of predictions that only have entities in specific types from instructions. Subscript scores are the deviation under three different rephrased instructions for each sample.

| Categories | Manually Designed Templates | Rephrased Templates | #Rephrased Templates |
|---|---|---|---|
| Default | Extract entities. | Identify the entities present in the text. | 219 |
| Base Type | Extract entities in types {types}. | Please identify the entities falling under the categories {types}. | 48 |
| Abstract Type | Extract entities in types {types}. | Please identify the entities falling under the categories {types}. | 48 |
| Description | Extract entities with following descriptions: {descriptions}. | Can you identify the entities described as followed: {descriptions}? | 104 |
| Importance | Extract the most important {num} entities. | Retrieve the {number} most essential entities. | 62 |
| Number | Extract {num} entities. | Fetch {number} entities. | 49 |
| Number+Base Type | Extract {num} entities in types {types}. | Could you identify {number} entities belonging to {types}? | 117 |
| Number+Abstract Type | Extract {num} entities in types {types}. | Retrieve {number} entities belonging to {types}. | 117 |

Table 8: Details of Instructions. {num}, {types}, {descriptions} are placeholders for entity numbers, types, and descriptions. We only display plural forms of templates in this table, but the templates will be different for singular or plural. We only show one example of rephrased templates for each category.

| Split | Corpus | | | Ontology | | | | Entity Info Density | | |
|---|---|---|---|---|---|---|---|---|---|---|
| | #Article | #Mention | #Triplets | #Ent. | #Aliases | #Rel. | #Types | %Desc. | %Aliases | %Types |
| Train | 11,447,454 | 39,930,663 | 19,184,948 | 2,234,052 | 840,401 | 962 | 21,350 | 93.5 | 64.2 | 71.5 |
| Dev | 2,710 | 13,601 unseen:3038 | 5,915 | 6,868 unseen:1417 | 8,812 | 234 | 1,163 | 94.6 | 55.6 | 70.2 |
| Test | 24,393 | 109,411 unseen:29,612 | 45,758 | 39,086 unseen:9,474 | 37,809 | 398 | 3,306 | 92.7 | 52.6 | 70.7 |

Table 9: Statistics of the instruction-following open-world dataset INSTRUCTOPENWIKI. We report the statistics of the corpus, including the number of articles, mentions, and triplets in the left section. The ontology statistics, including the number of unique entities, aliases, relations, and types, are reported in the middle section. The right section shows the proportions of mentions with descriptions, aliases, and types, respectively. Ent., Rel., and Desc. are short for entities, relations, and descriptions.

| Method | Partition | Instruction | MD F1 | EL F1(T=1) | EL F1(T=0.8) | ET F1 | OpenRE F1(CaRB) | EIG (Desc.) F1(ROUGE-L) | EIG (Aliases) F1 |
|---|---|---|---|---|---|---|---|---|---|
| ChatGPT | unseen | Base Type | $56.3_{1.0}$ | $31.4_{1.9}$ | $34.5_{2.3}$ | $16.5_{0.5}$ | $23.3_{0.9}$ | $41.4_{0.5}$ | $17.9_{1.2}$ |
| | | Abstract Type | $47.2_{0.6}$ | $23.7_{0.9}$ | $24.6_{0.9}$ | $10.3_{0.4}$ | $23.0_{1.6}$ | $39.5_{2.1}$ | $19.7_{0.4}$ |
| | | Description | $51.8_{0.1}$ | $24.5_{0.4}$ | $25.3_{0.3}$ | $3.3_{0.1}$ | $22.0_{0.0}$ | $56.3_{0.2}$ | $14.7_{0.6}$ |
| | | Macro Avg. | $51.8_{0.5}$ | $26.5_{0.5}$ | $28.1_{0.8}$ | $10.0_{0.3}$ | $22.8_{0.6}$ | $45.8_{0.6}$ | $17.4_{0.4}$ |
| | seen | Base Type | $50.1_{0.5}$ | $26.4_{0.9}$ | $27.4_{0.9}$ | $15.3_{0.6}$ | $21.0_{0.8}$ | $37.8_{1.1}$ | $13.6_{0.6}$ |
| | | Abstract Type | $47.5_{0.2}$ | $18.1_{0.8}$ | $19.3_{1.0}$ | $7.6_{0.5}$ | $21.0_{0.0}$ | $36.6_{0.8}$ | $16.7_{1.4}$ |
| | | Description | $46.6_{0.8}$ | $21.6_{1.0}$ | $22.9_{1.2}$ | $1.7_{0.3}$ | $22.0_{0.8}$ | $45.1_{1.3}$ | $25.0_{3.8}$ |
| | | Macro Avg. | $48.1_{0.2}$ | $22.1_{0.7}$ | $23.2_{0.8}$ | $8.2_{0.4}$ | $21.3_{0.5}$ | $39.8_{1.0}$ | $18.4_{1.6}$ |
| ChatGPT w/Demo | unseen | Base Type | $58.4_{0.7}$ | $42.0_{0.4}$ | $45.5_{0.3}$ | $30.0_{0.2}$ | $24.0_{0.0}$ | $45.8_{0.6}$ | $17.4_{1.4}$ |
| | | Abstract Type | $47.3_{0.0}$ | $37.6_{0.8}$ | $39.2_{0.9}$ | $16.8_{0.4}$ | $21.5_{1.5}$ | $51.9_{1.4}$ | $25.0_{2.7}$ |
| | | Description | $52.2_{0.1}$ | $39.2_{0.2}$ | $40.0_{0.3}$ | $6.7_{0.1}$ | $17.5_{0.5}$ | $83.5_{0.3}$ | $24.4_{0.5}$ |
| | | Macro Avg. | $52.6_{0.2}$ | $39.6_{0.3}$ | $41.6_{0.3}$ | $17.8_{0.2}$ | $21.0_{0.3}$ | $60.4_{0.8}$ | $22.3_{0.3}$ |
| | seen | Base Type | $55.1_{0.4}$ | $40.5_{0.6}$ | $41.5_{0.5}$ | $32.9_{0.3}$ | $21.0_{0.0}$ | $47.7_{0.1}$ | $12.6_{0.6}$ |
| | | Abstract Type | $49.1_{0.3}$ | $37.0_{0.2}$ | $38.4_{0.2}$ | $17.2_{0.2}$ | $21.0_{1.0}$ | $52.6_{0.6}$ | $18.7_{0.3}$ |
| | | Description | $47.6_{0.8}$ | $34.1_{1.4}$ | $35.4_{1.5}$ | $4.6_{0.1}$ | $17.0_{0.0}$ | $78.6_{0.3}$ | $38.7_{1.2}$ |
| | | Macro Avg. | $50.6_{0.2}$ | $37.2_{0.3}$ | $38.4_{0.4}$ | $18.2_{0.2}$ | $19.7_{0.3}$ | $59.7_{0.2}$ | $23.4_{0.7}$ |
| PIVOINE-1b | unseen | Base Type | $62.7_{0.5}$ | $51.3_{0.4}$ | $52.8_{0.4}$ | $38.3_{0.1}$ | $62.7_{0.5}$ | $64.7_{0.5}$ | $74.1_{0.1}$ |
| | | Abstract Type | $54.0_{0.2}$ | $44.6_{0.5}$ | $45.2_{0.5}$ | $32.2_{0.5}$ | $53.0_{1.4}$ | $68.7_{1.3}$ | $70.0_{1.0}$ |
| | | Description | $62.6_{0.4}$ | $50.6_{0.3}$ | $51.4_{0.3}$ | $44.6_{0.2}$ | $56.7_{0.5}$ | $75.1_{0.1}$ | $70.3_{0.1}$ |
| | | Macro Avg. | $59.8_{0.3}$ | $48.8_{0.2}$ | $49.8_{0.3}$ | $38.3_{0.2}$ | $57.4_{0.6}$ | $69.5_{0.5}$ | $71.5_{0.3}$ |
| | seen | Base Type | $61.7_{0.2}$ | $50.1_{0.1}$ | $51.0_{0.2}$ | $44.0_{0.1}$ | $56.3_{0.5}$ | $64.2_{0.1}$ | $70.9_{0.2}$ |
| | | Abstract Type | $59.0_{0.1}$ | $48.5_{0.2}$ | $49.3_{0.2}$ | $40.8_{0.1}$ | $56.0_{0.8}$ | $62.0_{0.2}$ | $70.1_{0.2}$ |
| | | Description | $67.8_{0.8}$ | $55.3_{0.6}$ | $57.1_{0.8}$ | $42.3_{0.2}$ | $62.7_{1.7}$ | $90.6_{0.0}$ | $75.0_{0.7}$ |
| | | Macro Avg. | $62.8_{0.3}$ | $51.3_{0.2}$ | $52.4_{0.3}$ | $42.4_{0.1}$ | $58.3_{0.8}$ | $72.3_{0.0}$ | $72.0_{0.3}$ |
| PIVOINE-7b | unseen | Base Type | $84.1_{0.4}$ | $74.4_{0.5}$ | $75.6_{0.6}$ | $56.0_{0.3}$ | $69.0_{0.8}$ | $74.1_{0.2}$ | $76.7_{0.1}$ |
| | | Abstract Type | $74.5_{0.2}$ | $65.0_{0.2}$ | $65.5_{0.1}$ | $52.0_{0.3}$ | $66.3_{0.5}$ | $78.8_{0.1}$ | $80.6_{0.1}$ |
| | | Description | $88.3_{0.1}$ | $75.9_{0.1}$ | $77.0_{0.1}$ | $64.0_{0.1}$ | $72.3_{0.5}$ | $87.0_{0.0}$ | $79.4_{0.1}$ |
| | | Macro Avg. | $82.3_{0.2}$ | $71.7_{0.2}$ | $72.7_{0.2}$ | $57.3_{0.2}$ | $69.2_{0.2}$ | $80.0_{0.0}$ | $78.9_{0.1}$ |
| | seen | Base Type | $76.3_{0.2}$ | $66.7_{0.1}$ | $67.5_{0.1}$ | $59.2_{0.2}$ | $65.7_{0.9}$ | $76.9_{0.2}$ | $78.6_{0.0}$ |
| | | Abstract Type | $78.5_{0.3}$ | $69.8_{0.2}$ | $70.4_{0.2}$ | $59.6_{0.1}$ | $69.7_{0.5}$ | $76.8_{0.1}$ | $80.0_{0.1}$ |
| | | Description | $90.5_{0.1}$ | $80.9_{0.1}$ | $81.4_{0.1}$ | $59.6_{0.3}$ | $80.0_{0.0}$ | $97.0_{0.0}$ | $81.3_{0.0}$ |
| | | Macro Avg. | $81.8_{0.1}$ | $72.5_{0.1}$ | $73.1_{0.1}$ | $59.5_{0.1}$ | $71.8_{0.4}$ | $83.6_{0.1}$ | $80.0_{0.0}$ |

Table 10: Main results of the instruction generalization evaluation on three instruction categories "Base Type", "Abstract Type", and "Description". Headers are the same as Tab. 1. Partition denotes the unseen and seen instructions. Subscript scores are the deviation under three different rephrased instructions for each sample.

| Instruction | GENRE P | R | F1 | ChatGPT P | R | F1 | ChatGPT w/ Demo P | R | F1 | PIVOINE-1b P | R | F1 | PIVOINE-7b P | R | F1 |
|---|---|---|---|---|---|---|---|---|---|---|---|---|---|---|---|
| Default | 38.8 | 50.0 | 43.7 | $53.4_{0.1}$ | $57.5_{0.2}$ | $55.4_{0.1}$ | $51.2_{0.1}$ | $51.9_{0.1}$ | $51.6_{0.1}$ | $71.1_{0.0}$ | $57.3_{0.0}$ | $63.5_{0.0}$ | $82.0_{0.0}$ | $76.4_{0.0}$ | $79.1_{0.0}$ |
| Base Type | -- | -- | -- | $41.9_{0.4}$ | $66.3_{1.0}$ | $51.4_{0.6}$ | $47.9_{0.1}$ | $66.8_{0.3}$ | $55.8_{0.2}$ | $65.8_{0.1}$ | $58.5_{0.0}$ | $61.9_{0.1}$ | $76.0_{0.2}$ | $80.2_{0.1}$ | $78.1_{0.1}$ |
| Abstract Type | -- | -- | -- | $38.2_{0.4}$ | $62.5_{0.2}$ | $47.4_{0.3}$ | $42.5_{0.3}$ | $57.1_{0.1}$ | $48.7_{0.2}$ | $61.2_{0.1}$ | $54.9_{0.2}$ | $57.9_{0.1}$ | $76.8_{0.1}$ | $78.3_{0.2}$ | $77.5_{0.2}$ |
| Description | -- | -- | -- | $43.9_{0.2}$ | $61.1_{0.4}$ | $51.1_{0.0}$ | $44.7_{0.0}$ | $60.9_{0.2}$ | $51.6_{0.1}$ | $65.9_{0.1}$ | $60.8_{0.6}$ | $63.2_{0.3}$ | $88.7_{0.1}$ | $88.4_{0.1}$ | $88.5_{0.1}$ |
| Importance | -- | -- | -- | $46.8_{0.5}$ | $47.9_{0.2}$ | $47.4_{0.3}$ | $47.5_{0.4}$ | $48.2_{0.1}$ | $47.8_{0.1}$ | $64.1_{0.1}$ | $57.6_{0.5}$ | $60.7_{0.3}$ | $74.8_{0.2}$ | $74.7_{0.1}$ | $74.7_{0.1}$ |
| Macro Avg. | -- | -- | -- | $44.9_{0.2}$ | $59.1_{0.1}$ | $50.5_{0.2}$ | $46.8_{0.1}$ | $57.0_{0.0}$ | $51.1_{0.1}$ | $65.6_{0.0}$ | $57.8_{0.0}$ | $61.4_{0.0}$ | $79.7_{0.1}$ | $79.6_{0.0}$ | $79.6_{0.0}$ |

Table 11: Main results of mention detection (MD). We report precision, recall, and F1 score of each model in this table. The subscript scores are the deviation under three different rephrased instructions for each instruction category. Subscript scores are the deviation under three different rephrased instructions for each sample. GENRE is an closed-world generative IE baseline without the instruction-following ability, so we only report GENRE scores with the default instruction.

| Instruction | GENRE F1(T=1) | F1(T=0.8) | ChatGPT F1(T=1) | F1(T=0.8) | ChatGPT w/Demo F1(R=1) | F1(R=0.8) | PIVOINE-1b F1(R=1) | F1(R=0.8) | PIVOINE-7b F1(R=1) | F1(R=0.8) |
|---|---|---|---|---|---|---|---|---|---|---|
| Default | 17.2 | 20.1 | $27.4_{0.1}$ | $28.3_{0.0}$ | $39.6_{0.0}$ | $41.0_{0.0}$ | $50.3_{0.0}$ | $51.3_{0.0}$ | $69.4_{0.1}$ | $70.1_{0.0}$ |
| Base Type | —— | —— | $27.4_{0.4}$ | $28.8_{0.4}$ | $40.8_{0.4}$ | $42.3_{0.4}$ | $50.4_{0.0}$ | $51.4_{0.1}$ | $68.5_{0.1}$ | $69.3_{0.1}$ |
| Abstract Type | —— | —— | $19.3_{0.5}$ | $20.4_{0.6}$ | $37.1_{0.3}$ | $38.6_{0.3}$ | $47.6_{0.2}$ | $48.4_{0.2}$ | $68.7_{0.2}$ | $69.2_{0.2}$ |
| Description | —— | —— | $24.2_{0.3}$ | $24.9_{0.1}$ | $38.5_{0.0}$ | $39.4_{0.0}$ | $51.2_{0.2}$ | $52.1_{0.2}$ | $76.4_{0.1}$ | $77.5_{0.1}$ |
| Importance | —— | —— | $26.7_{0.4}$ | $28.0_{0.4}$ | $36.8_{0.1}$ | $38.2_{0.0}$ | $48.7_{0.2}$ | $49.8_{0.3}$ | $66.2_{0.1}$ | $67.2_{0.1}$ |
| Macro Avg. | —— | —— | $25.0_{0.2}$ | $26.1_{0.2}$ | $38.6_{0.0}$ | $39.9_{0.0}$ | $49.6_{0.0}$ | $50.6_{0.0}$ | $69.8_{0.1}$ | $70.7_{0.1}$ |

Table 12: Main results of end-to-end entity linking (EL). We report F1 score calculated by title matching with ROUGE-L F1 threshold 1 and 0.8 in this table. Subscript scores are the deviation under three different rephrased instructions for each sample. GENRE is an closed-world generative IE baseline without the instruction-following ability, so we only report GENRE scores with the default instruction.

| Partition | Instruction | GENRE | ChatGPT | ChatGPT w/Demo | PIVOINE-1 | PIVOINE-7b |
|---|---|---|---|---|---|---|
| Before 05/30/2022 | Default | 55.3 | $56.5_{0.2}$ | $50.1_{0.2}$ | $56.8_{0.0}$ | $77.8_{0.0}$ |
| | Base Type | —— | $66.1_{1.2}$ | $65.4_{0.5}$ | $59.4_{0.2}$ | $86.5_{0.1}$ |
| | Abstract Type | —— | $63.6_{0.7}$ | $56.7_{0.1}$ | $56.4_{0.1}$ | $82.9_{0.1}$ |
| | Description | —— | $60.9_{0.3}$ | $61.1_{0.2}$ | $62.8_{0.8}$ | $90.4_{0.1}$ |
| | Importance | —— | $49.1_{0.1}$ | $49.0_{0.1}$ | $60.7_{0.6}$ | $80.7_{0.1}$ |
| | Macro Avg. | —— | $59.3_{0.3}$ | $56.5_{0.0}$ | $59.2_{0.1}$ | $83.7_{0.0}$ |
| After 05/30/2022 | Default | 33.6 | $60.5_{0.1}$ | $57.4_{0.2}$ | $58.7_{0.0}$ | $71.8_{0.0}$ |
| | Base Type | —— | $66.7_{1.1}$ | $69.0_{0.0}$ | $57.1_{0.2}$ | $70.4_{0.4}$ |
| | Abstract Type | —— | $60.6_{0.8}$ | $57.8_{0.0}$ | $52.5_{0.4}$ | $70.9_{0.5}$ |
| | Description | —— | $61.5_{0.6}$ | $60.5_{0.2}$ | $56.5_{0.2}$ | $84.1_{0.2}$ |
| | Importance | —— | $45.3_{0.5}$ | $46.4_{0.1}$ | $50.9_{0.5}$ | $61.6_{0.3}$ |
| | Macro Avg. | —— | $58.9_{0.2}$ | $58.2_{0.0}$ | $55.1_{0.0}$ | $71.7_{0.1}$ |

Table 13: Results of generalization study on mention detection (MD). We report recalls of mention spans in this table. The partition "Before 05/30/2022" denotes mentions linked to the training ontology (seen entities), while "After 05/30/2022" denotes mentions linked to entities that are out of training ontology (unseen entities). Subscript scores are the deviation under three different rephrased instructions for each sample. GENRE is an closed-world generative IE baseline without the instruction-following ability, so we only report GENRE scores with the default instruction.

| Partition | Instruction | GENRE R(T=1) | R(T=0.8) | ChatGPT R(T=1) | R(T=0.8) | ChatGPT w/Demo R(T=1) | R(T=0.8) | PIVOINE-1 R(T=1) | R(T=0.8) | PIVOINE-7b R(T=1) | R(T=0.8) |
|---|---|---|---|---|---|---|---|---|---|---|---|
| Before 05/30/2022 | Default | 23.8 | 24.7 | $29.6_{0.1}$ | $30.2_{0.1}$ | $40.2_{0.1}$ | $41.0_{0.1}$ | $48.5_{0.0}$ | $49.0_{0.0}$ | $73.0_{0.0}$ | $73.3_{0.0}$ |
| | Base Type | —— | —— | $39.3_{0.5}$ | $39.7_{0.4}$ | $50.4_{0.9}$ | $51.0_{0.9}$ | $52.9_{0.0}$ | $53.0_{0.0}$ | $81.6_{0.1}$ | $81.8_{0.1}$ |
| | Abstract Type | —— | —— | $29.5_{0.9}$ | $29.9_{0.9}$ | $47.8_{0.2}$ | $48.3_{0.2}$ | $49.5_{0.1}$ | $49.8_{0.1}$ | $78.0_{0.3}$ | $78.1_{0.3}$ |
| | Description | —— | —— | $30.9_{0.7}$ | $31.2_{0.7}$ | $47.7_{0.1}$ | $48.3_{0.1}$ | $55.5_{0.6}$ | $55.8_{0.6}$ | $85.3_{0.2}$ | $85.7_{0.2}$ |
| | Importance | —— | —— | $29.8_{0.6}$ | $30.4_{0.6}$ | $39.1_{0.2}$ | $39.9_{0.1}$ | $52.7_{0.5}$ | $53.3_{0.5}$ | $76.3_{0.0}$ | $76.6_{0.0}$ |
| | Macro Avg. | —— | —— | $31.8_{0.2}$ | $32.3_{0.2}$ | $45.0_{0.1}$ | $45.7_{0.1}$ | $51.8_{0.0}$ | $52.2_{0.0}$ | $78.8_{0.1}$ | $79.1_{0.1}$ |
| After 05/30/2022 | Default | 0 | 3.6 | $24.7_{0.1}$ | $27.1_{0.1}$ | $38.7_{0.2}$ | $41.8_{0.2}$ | $35.4_{0.0}$ | $37.8_{0.0}$ | $48.3_{0.1}$ | $50.4_{0.1}$ |
| | Base Type | —— | —— | $29.3_{0.9}$ | $33.1_{1.1}$ | $46.5_{0.1}$ | $50.0_{0.2}$ | $39.2_{0.1}$ | $41.4_{0.1}$ | $52.5_{0.6}$ | $54.6_{0.7}$ |
| | Abstract Type | —— | —— | $18.9_{1.0}$ | $21.9_{1.0}$ | $36.8_{0.2}$ | $40.3_{0.3}$ | $38.3_{0.4}$ | $39.7_{0.4}$ | $55.5_{0.4}$ | $56.8_{0.4}$ |
| | Description | —— | —— | $24.8_{0.9}$ | $27.0_{0.9}$ | $40.7_{0.0}$ | $42.9_{0.0}$ | $36.4_{0.2}$ | $38.3_{0.2}$ | $57.8_{0.1}$ | $60.3_{0.1}$ |
| | Importance | —— | —— | $21.3_{0.1}$ | $23.6_{0.3}$ | $32.8_{0.4}$ | $35.4_{0.3}$ | $32.3_{0.2}$ | $34.3_{0.3}$ | $44.5_{0.3}$ | $46.8_{0.2}$ |
| | Macro Avg. | —— | —— | $23.8_{0.2}$ | $26.5_{0.3}$ | $39.1_{0.0}$ | $42.1_{0.0}$ | $36.3_{0.1}$ | $38.3_{0.1}$ | $51.7_{0.2}$ | $53.8_{0.2}$ |

Table 14: Results of generalization study on end-to-end entity linking (EL). We report recalls calculated by title matching with ROUGE-L F1 threshold 1 and 0.8 in this table. The partition "Before 05/30/2022" denotes mentions linked to the training ontology (seen entities), while "After 05/30/2022" denotes mentions linked to entities that are out of training ontology (unseen entities). Subscript scores are the deviation under three different rephrased instructions for each sample. GENRE is an closed-world generative IE baseline without the instruction-following ability, so we only report GENRE scores with the default instruction.

| Instruction | OpenIE6 | | | ChatGPT | | | ChatGPT w/Demo | | | PIVOINE-1b | | | PIVOINE-7b | | |
|---|---|---|---|---|---|---|---|---|---|---|---|---|---|---|---|
| | P | R | F1 | P | R | F1 | P | R | F1 | P | R | F1 | P | R | F1 |
| Default | 13.0 | 18.5 | 15.2 | $17.8_{0.0}$ | $36.9_{0.1}$ | $23.7_{0.5}$ | $16.4_{0.0}$ | $37.8_{0.2}$ | $22.0_{0.0}$ | $54.1_{0.0}$ | $55.8_{0.0}$ | $54.0_{0.0}$ | $65.7_{0.1}$ | $68.1_{0.0}$ | $66.3_{0.5}$ |
| Base Type | — — | — — | — — | $16.1_{0.3}$ | $34.6_{1.3}$ | $21.7_{0.5}$ | $15.2_{0.1}$ | $43.0_{0.3}$ | $22.0_{0.0}$ | $56.6_{0.7}$ | $61.6_{0.5}$ | $58.3_{0.5}$ | $60.3_{1.2}$ | $75.7_{0.4}$ | $66.7_{0.9}$ |
| Abstract Type | — — | — — | — — | $15.7_{0.6}$ | $37.8_{0.3}$ | $22.0_{0.8}$ | $14.6_{0.4}$ | $46.4_{0.6}$ | $21.5_{0.5}$ | $56.7_{1.0}$ | $54.5_{0.1}$ | $55.0_{0.8}$ | $66.4_{0.3}$ | $71.5_{0.1}$ | $68.3_{0.5}$ |
| Description | — — | — — | — — | $16.6_{0.1}$ | $35.8_{0.3}$ | $22.0_{0.0}$ | $12.0_{0.2}$ | $35.2_{0.8}$ | $17.5_{0.5}$ | $52.3_{0.6}$ | $63.8_{0.6}$ | $57.0_{0.0}$ | $69.3_{0.4}$ | $77.7_{0.1}$ | $73.0_{0.0}$ |
| Importance | — — | — — | — — | $18.1_{0.2}$ | $35.8_{0.4}$ | $23.7_{0.5}$ | $17.2_{0.3}$ | $39.6_{0.1}$ | $23.5_{0.5}$ | $54.9_{0.3}$ | $57.9_{0.4}$ | $55.7_{0.5}$ | $62.4_{0.1}$ | $68.8_{0.2}$ | $65.0_{0.0}$ |
| Macro Avg. | — — | — — | — — | $16.9_{0.2}$ | $36.2_{0.2}$ | $22.6_{0.3}$ | $15.1_{0.1}$ | $40.4_{0.2}$ | $21.3_{0.1}$ | $54.9_{0.4}$ | $58.7_{0.2}$ | $56.0_{0.3}$ | $64.8_{0.3}$ | $72.4_{0.0}$ | $67.9_{0.2}$ |

Table 15: Main results of open relation extraction (Open RE). We report precision, recall, and F1 with the CaRB scoring based on the ROUGE-L matcher. Subscript scores are the deviation under three different rephrased instructions for each sample. OPENIE6 is an openIE baseline without the instruction-following ability, so we only report OpenIE6 scores with the default instruction.

| Partition | Instruction | OpenIE6 | ChatGPT | ChatGPT w/ Demo | PIVOINE-1b | PIVOINE-7b |
|---|---|---|---|---|---|---|
| Before 05/30/2022 | Default | 21.9 | $35.1_{0.1}$ | $35.9_{0.1}$ | $57.7_{0.0}$ | $73.3_{0.0}$ |
| | Base Type | — — | $32.1_{1.7}$ | $40.4_{0.8}$ | $63.6_{0.6}$ | $83.7_{0.5}$ |
| | Abstract Type | — — | $36.0_{0.6}$ | $43.7_{0.6}$ | $58.8_{0.1}$ | $78.4_{0.1}$ |
| | Description | — — | $33.3_{0.5}$ | $33.0_{0.7}$ | $67.9_{0.6}$ | $84.1_{0.1}$ |
| | Importance | — — | $34.2_{0.5}$ | $38.5_{0.2}$ | $61.0_{0.6}$ | $77.4_{0.2}$ |
| | Macro Avg. | — — | $34.1_{0.2}$ | $38.3_{0.0}$ | $61.8_{0.3}$ | $79.4_{0.1}$ |
| After 05/30/2022 | Default | 18.2 | $32.7_{0.2}$ | $34.0_{0.2}$ | $35.8_{0.1}$ | $36.1_{0.1}$ |
| | Base Type | — — | $23.6_{0.8}$ | $27.8_{0.6}$ | $39.5_{0.2}$ | $37.9_{0.2}$ |
| | Abstract Type | — — | $28.0_{1.1}$ | $33.5_{1.3}$ | $32.6_{0.0}$ | $36.4_{0.4}$ |
| | Description | — — | $24.8_{0.2}$ | $22.4_{0.5}$ | $30.3_{0.4}$ | $35.3_{0.2}$ |
| | Importance | — — | $26.2_{0.7}$ | $29.0_{0.6}$ | $34.1_{0.5}$ | $34.6_{0.1}$ |
| | Macro Avg. | — — | $27.1_{0.5}$ | $29.3_{0.5}$ | $34.5_{0.1}$ | $36.1_{0.1}$ |

Table 16: Results of generalization study on open relation extraction (Open RE). We report recall with the CaRB scoring based on the ROUGE-L matcher. The partition "Before 05/30/2022" denotes mentions linked to the training ontology (seen entities), while "After 05/30/2022" denotes mentions linked to entities that are out of training ontology (unseen entities). Subscript scores are the deviation under three different rephrased instructions for each sample. OPENIE6 is an openIE baseline without the instruction-following ability, so we only report OpenIE6 scores with the default instruction.

| Instruction | ChatGPT | | | ChatGPT w/ Demo | | | PIVOINE-1b | | | PIVOINE-7b | | |
|---|---|---|---|---|---|---|---|---|---|---|---|---|
| | P | R | F1 | P | R | F1 | P | R | F1 | P | R | F1 |
| Default | $2.7_{0.0}$ | $3.8_{0.0}$ | $3.1_{0.0}$ | $6.4_{0.0}$ | $7.5_{0.0}$ | $6.9_{0.0}$ | $46.3_{0.1}$ | $37.9_{0.1}$ | $41.7_{0.1}$ | $61.3_{0.0}$ | $55.3_{0.0}$ | $58.1_{0.0}$ |
| Base Type | $13.0_{0.3}$ | $19.4_{0.5}$ | $15.6_{0.4}$ | $29.9_{0.2}$ | $35.0_{0.4}$ | $32.3_{0.3}$ | $46.0_{0.1}$ | $39.9_{0.2}$ | $42.7_{0.1}$ | $59.3_{0.2}$ | $57.6_{0.1}$ | $58.4_{0.1}$ |
| Abstract Type | $8.9_{0.5}$ | $7.5_{0.2}$ | $8.2_{0.3}$ | $16.2_{0.3}$ | $18.2_{0.2}$ | $17.1_{0.3}$ | $42.7_{0.2}$ | $35.6_{0.0}$ | $38.8_{0.1}$ | $62.3_{0.3}$ | $53.9_{0.1}$ | $57.8_{0.1}$ |
| Description | $2.3_{0.1}$ | $4.4_{0.1}$ | $3.1_{0.1}$ | $6.4_{0.1}$ | $6.5_{0.1}$ | $6.4_{0.1}$ | $44.9_{0.1}$ | $43.8_{0.4}$ | $44.4_{0.1}$ | $64.1_{0.1}$ | $63.2_{0.1}$ | $63.6_{0.1}$ |
| Importance | $2.6_{0.2}$ | $3.8_{0.3}$ | $3.1_{0.2}$ | $5.9_{0.0}$ | $7.2_{0.1}$ | $6.5_{0.0}$ | $39.5_{0.2}$ | $39.5_{0.3}$ | $39.5_{0.1}$ | $50.6_{0.0}$ | $53.4_{0.1}$ | $52.0_{0.0}$ |
| Macro Avg. | $7.5_{0.1}$ | $8.8_{0.0}$ | $7.9_{0.0}$ | $14.4_{0.0}$ | $15.4_{0.2}$ | $14.8_{0.1}$ | $42.4_{0.0}$ | $38.7_{0.1}$ | $40.4_{0.0}$ | $57.7_{0.0}$ | $55.3_{0.1}$ | $56.4_{0.0}$ |

Table 17: Main results of fine-grained entity typing (ET). We report precision, recall, and F1 scores based on exact matching of type names in this table. Subscript scores are the deviation under three different rephrased instructions for each sample.

| Partition | Instruction | ChatGPT | ChatGPT w/Demo | PIVOINE-1b | PIVOINE-7b |
|---|---|---|---|---|---|
| Before 05/30/2022 | Default | $4.0_{0.0}$ | $7.8_{0.1}$ | $41.9_{0.1}$ | $63.4_{0.0}$ |
| | Base Type | $21.1_{0.4}$ | $34.4_{0.7}$ | $46.2_{0.2}$ | $71.3_{0.1}$ |
| | Abstract Type | $8.4_{0.2}$ | $19.2_{0.1}$ | $40.0_{0.0}$ | $65.6_{0.3}$ |
| | Description | $5.0_{0.1}$ | $7.6_{0.1}$ | $50.6_{0.5}$ | $73.3_{0.1}$ |
| | Importance | $4.2_{0.5}$ | $7.6_{0.3}$ | $47.1_{0.3}$ | $65.5_{0.1}$ |
| | Macro Avg. | $9.7_{0.1}$ | $15.9_{0.2}$ | $44.4_{0.1}$ | $66.8_{0.1}$ |
| After 05/30/2022 | Default | $2.6_{0.0}$ | $5.9_{0.3}$ | $17.7_{0.1}$ | $13.8_{0.0}$ |
| | Base Type | $16.1_{0.6}$ | $36.2_{0.0}$ | $27.2_{0.3}$ | $30.2_{0.0}$ |
| | Abstract Type | $5.8_{0.5}$ | $16.3_{0.3}$ | $27.4_{0.1}$ | $32.2_{0.3}$ |
| | Description | $2.7_{0.4}$ | $3.1_{0.2}$ | $21.0_{0.2}$ | $29.1_{0.1}$ |
| | Importance | $2.7_{0.4}$ | $6.0_{0.9}$ | $12.7_{0.1}$ | $10.7_{0.2}$ |
| | Macro Avg. | $6.9_{0.1}$ | $14.2_{0.1}$ | $22.8_{0.0}$ | $24.1_{0.2}$ |

Table 18: Results of generalization study on entity typing (ET). We report recalls of entity types based on the exact matching of type names. The partition "Before 05/30/2022" denotes mentions linked to the training ontology (seen entities), while "After 05/30/2022" denotes mentions linked to entities that are out of training ontology (unseen entities). Subscript scores are the deviation under three different rephrased instructions for each sample.

| Partition | Instruction | ChatGPT | ChatGPT w/Demo | PIVOINE-1b | PIVOINE-7b |
|---|---|---|---|---|---|
| Before 05/30/2022 | Default | $36.2_{0.2}$ | $46.3_{0.1}$ | $74.3_{0.0}$ | $83.4_{0.0}$ |
| | Base Type | $38.9_{0.6}$ | $48.7_{0.2}$ | $67.4_{0.1}$ | $79.5_{0.1}$ |
| | Abstract Type | $40.6_{1.5}$ | $56.2_{1.4}$ | $67.5_{0.4}$ | $81.3_{0.1}$ |
| | Description | $57.8_{0.5}$ | $85.9_{0.6}$ | $75.8_{0.3}$ | $86.8_{0.1}$ |
| | Importance | $38.2_{0.5}$ | $46.5_{0.1}$ | $75.9_{0.2}$ | $84.6_{0.1}$ |
| | Macro Avg. | $40.2_{0.4}$ | $53.6_{0.2}$ | $71.4_{0.1}$ | $82.5_{0.0}$ |
| After 05/30/2022 | Default | $36.8_{0.3}$ | $45.0_{0.2}$ | $51.6_{0.1}$ | $60.7_{0.1}$ |
| | Base Type | $37.9_{0.9}$ | $44.6_{1.1}$ | $54.7_{0.4}$ | $60.0_{0.3}$ |
| | Abstract Type | $28.7_{0.3}$ | $44.1_{0.8}$ | $51.3_{0.4}$ | $58.8_{0.0}$ |
| | Description | $47.7_{0.7}$ | $75.9_{0.6}$ | $81.5_{0.6}$ | $93.1_{0.2}$ |
| | Importance | $36.0_{1.1}$ | $44.1_{0.8}$ | $45.0_{0.8}$ | $57.9_{0.3}$ |
| | Macro Avg. | $36.0_{0.3}$ | $48.1_{0.2}$ | $55.7_{0.3}$ | $64.1_{0.2}$ |

Table 19: Results of generalization study on entity description generation. We report average ROUGE-L F1 scores in this table. The partition "Before 05/30/2022" denotes mentions linked to the training ontology (seen entities), while "After 05/30/2022" denotes mentions linked to entities that are out of training ontology (unseen entities). Subscript scores are the deviation under three different rephrased instructions for each sample.

| Partition | Instruction | ChatGPT | ChatGPT w/Demo | PIVOINE-1b | PIVOINE-7b |
|---|---|---|---|---|---|
| Before 05/30/2022 | Default | $8.5_{0.1}$ | $1.1_{0.0}$ | $69.8_{0.0}$ | $78.0_{0.0}$ |
| | Base Type | $9.1_{0.4}$ | $7.7_{0.2}$ | $71.3_{0.1}$ | $76.7_{0.1}$ |
| | Abstract Type | $10.1_{0.4}$ | $14.8_{0.2}$ | $67.3_{0.4}$ | $78.7_{0.2}$ |
| | Description | $10.1_{0.4}$ | $19.4_{0.5}$ | $65.6_{0.2}$ | $76.2_{0.1}$ |
| | Importance | $10.1_{0.6}$ | $1.1_{0.3}$ | $75.1_{0.1}$ | $81.6_{0.1}$ |
| | Macro Avg. | $10.1_{0.1}$ | $8.4_{0.0}$ | $69.7_{0.1}$ | $78.0_{0.0}$ |
| After 05/30/2022 | Default | $10.9_{0.6}$ | $1.8_{0.1}$ | $16.9_{0.1}$ | $21.1_{0.1}$ |
| | Base Type | $14.6_{1.3}$ | $11.9_{1.8}$ | $22.7_{0.3}$ | $24.5_{0.5}$ |
| | Abstract Type | $17.1_{2.5}$ | $21.4_{1.3}$ | $23.6_{0.2}$ | $27.5_{0.3}$ |
| | Description | $15.7_{1.6}$ | $37.4_{1.2}$ | $18.9_{0.1}$ | $17.9_{0.3}$ |
| | Importance | $14.0_{1.4}$ | $1.8_{0.2}$ | $24.5_{0.2}$ | $28.7_{0.2}$ |
| | Macro Avg. | $14.3_{0.3}$ | $13.3_{0.1}$ | $20.5_{0.2}$ | $22.7_{0.1}$ |

Table 20: Results of generalization study on entity aliases generation. We report recalls of entity aliases based on exact matching in this table. The partition "Before 05/30/2022" denotes mentions linked to the training ontology (seen entities), while "After 05/30/2022" denotes mentions linked to entities that are out of training ontology (unseen entities). Subscript scores are the deviation under three different rephrased instructions for each sample.

| Instruction | ChatGPT | | | ChatGPT w/Demo | | | PIVOINE-1b | | | PIVOINE-7b | | |
|---|---|---|---|---|---|---|---|---|---|---|---|---|
| | MD | IE(T=1) | IE(T=0.8) | MD | IE(T=1) | IE(T=0.8) | MD | IE(T=1) | IE(T=0.8) | MD | IE(T=1) | IE(T=0.8) |
| Number | $51.4_{0.1}$ | $24.8_{0.6}$ | $25.9_{0.7}$ | $49.0_{0.1}$ | $37.5_{0.2}$ | $39.1_{0.2}$ | $69.9_{0.1}$ | $55.9_{0.1}$ | $57.1_{0.1}$ | $81.3_{0.1}$ | $71.3_{0.1}$ | $72.2_{0.1}$ |
| Number+ Base Type | $53.1_{0.3}$ | $25.8_{0.3}$ | $27.1_{0.2}$ | $50.3_{0.0}$ | $38.4_{0.1}$ | $40.1_{0.0}$ | $69.4_{0.2}$ | $55.2_{0.2}$ | $56.9_{0.2}$ | $80.8_{0.1}$ | $71.2_{0.1}$ | $72.2_{0.1}$ |
| Number+ Abstract Type | $53.3_{0.5}$ | $25.0_{0.2}$ | $26.2_{0.1}$ | $51.1_{0.1}$ | $38.0_{0.1}$ | $39.5_{0.1}$ | $72.4_{0.3}$ | $57.4_{0.2}$ | $59.0_{0.2}$ | $82.0_{0.1}$ | $73.6_{0.2}$ | $74.3_{0.2}$ |
| Macro Avg. | $52.6_{0.3}$ | $25.2_{0.1}$ | $26.4_{0.2}$ | $50.1_{0.0}$ | $38.0_{0.1}$ | $39.6_{0.1}$ | $70.6_{0.1}$ | $56.2_{0.0}$ | $57.7_{0.0}$ | $81.4_{0.1}$ | $72.0_{0.1}$ | $72.9_{0.1}$ |

Table 21: Results of partial extraction instructions on mention detection (MD) and information extraction (IE). We report precision for mention spans in MD and title matching with ROUGE-L F1 threshold 0.8 and 1 in IE. The partition "Before 05/30/2022" denotes mentions linked to the training ontology (seen entities), while "After 05/30/2022" denotes mentions linked to entities that are out of training ontology (unseen entities). Subscript scores are the deviation under three different rephrased instructions for each sample.

| Category | Prompt |
|---|---|
| Default | Context: "Extract entities."\n\n Please rephrase this context. |
| Base Type | Context: "Extract entities in types {types}."\n\n {types} in the context is a placeholder for a list of entity types. Please rephrase this context and keep {types} in the rephrased sentence. {types} should be put after the word "types"" |
| Abstract Type | Context: "Extract entities in types {types}."\n\n {types} in the context is a placeholder for a list of entity types. Please rephrase this context and keep {types} in the rephrased sentence. {types} should be put after the word "types"" |
| Description | Context: "Extract entities in following descriptions: {descriptions}"\n\n {descriptions} in the context is a placeholder for a list of entity descriptions. Please rephrase this context and keep {descriptions} in the rephrased sentence. |
| Importance | Context: "Extract the most important {number} entities."\n\n {number} in the context is a placeholder for the number of entity. Please rephrase this context and keep {number} in the rephrased sentence. |
| Number | Context: "Extract {number} entities."\n\n {number} in the context is a placeholder for the number of entity. Please rephrase this context and keep {number} in the rephrased sentence. |
| Number+Base Type | Context: "Extract {number} entities in types {types}."\n\n {types} in the context is a placeholder for a list of entity types. {number} in the context is a placeholder for the number of entities. Please rephrase this context and keep {types} and {number} in the rephrased sentence. {types} should be put after the word "types"" |
| Number+Abstract Type | Context: "Extract {number} entities in types {types}."\n\n {types} in the context is a placeholder for a list of entity types. {number} in the context is a placeholder for the number of entities. Please rephrase this context and keep {types} and {number} in the rephrased sentence. {types} should be put after the word "types"" |

Table 22: Details of ChatGPT prompts we used to rephrase manually designed templates.

| Type | Prompt |
|---|---|
| ChatGPT | [context]: {context}.\n[instruction]: {instruction}.\n\nPlease provide the response in the JSON format. The response should contains entities and triplets. Each entity has its mention, title, a list of types, description, and a list of aliases. Each triplet has its head and tail mentions, and a list of relations. Here is an example of the return JSON format: {"entities": [{"mention": String, "title": String, "type": List[String], "description": String, "aliases":List[String]}], "triplets": [{"head": String, "tail": String, "relations": List[String]}]}. |

Table 23: Details of ChatGPT prompts we used to address instruction-following open-world IE. {context}, {instruction} are placeholders for input context, instruction, and one-shot example, respectively.