# OpenReview forum: "PIVOINE: Instruction Tuning for Open-world Entity Profiling"
_EMNLP/2023/Conference — EMNLP 2023 Findings_

### Official Review · Reviewer_VgsG · 2023-08-04

**Typos Grammar Style And Presentation Improvements:** 1. In line 182, there appears to be a…
**Soundness:** 3

**Excitement:**

4: Strong: This paper deepens the understanding of some phenomenon or lowers the barriers to an existing research direction.

**Paper Topic And Main Contributions:**

The paper addresses the challenge of Open World Information Extraction (IE) through instruction tuning, presenting a new concept of generating comprehensive entity profiles. Unlike previous work that focuses on specific tasks (e.g., NER, entity linking) within IE, this paper integrates six relevant tasks to create entity profiles. The primary contributions of the paper include the development of the INSTRUCTOPENWIKI, a large-scale instruction-following open-world IE dataset, and the creation of the PIVOINE model, an instruction tuning model based on BLOOM.

**Questions For The Authors:**

A. How is the evaluation of cross-instructions conducted? The paper shows that the Number + Base Type and Number + Abstract Type instructions have high error rates (Fig 4). Could you clarify what is considered wrong in these cases? Is it the extracted numbers or the types?

B. Given the relation extraction data are created by aligning Wikipedia with Wikidata with distant supervision, some concerns about the dataset were raised. In prior work, Gao et al., (2021) found that distant supervision often brings a lot of noise and causes wrong relations labels. In addition, a pair of head and tail entities might also be related with multiple relations. Can you elaborate on how the relation is being selected and how the dataset's quality is being ensured?

Gao et al., (2021) : Manual Evaluation Matters: Reviewing Test Protocols of Distantly Supervised Relation Extraction

C: The concept of generating entity profiles is interesting,  however, the evaluations are still conducted using single evaluation metrics, with no comprehensive evaluation of each entity. Is it possible to provide overall score for on entity level?

**Reasons To Accept:**

This is the first work in Open-world IE that demonstrates the ability of large language models to be customized to generate complete entity profiles, as opposed to focusing on single tasks. This approach will facilitate future research in the direction of customized and complete OpenIE.

The authors introduce the INSTRUCTOPENWIKI dataset and the PIVOINE model, providing valuable resources for future research in this area. They conduct extensive experiments that offer a comprehensive evaluation of the model's performance, demonstrating its effectiveness in following instructions and extracting entity profiles.

The authors propose a delicate method that uses the time difference between Wiki dumps to construct a strictly open-world test set. This method minimizes potential data leakage and maximizes the completeness of the training set’s annotations.

**Reasons To Reject:**

While the authors claim that the model can follow unseen instructions, the definition of unseen instructions seems to primarily focus on unseen entities, and the model fails to follow cross-instructions, which are also unseen. The extent of what is considered unseen is debatable. For example, the "2023 ATP Tour" example in Figure 1, although it doesn’t occur in the training data, “2022 ATP Tour” is included in the training data. This raises concerns that the model may simply mimic what’s encoded in the training data, making the comparison against models that are untrained (e.g., ChatGPT) unfair.

**Reproducibility:**

4: Could mostly reproduce the results, but there may be some variation because of sample variance or minor variations in their interpretation of the protocol or method.

**Reviewer Confidence:**

4: Quite sure. I tried to check the important points carefully. It's unlikely, though conceivable, that I missed something that should affect my ratings.

---

> ### Author Rebuttal · Authors · 2023-08-29
>
> We thank the reviewer for great questions. We first would like to clarify the concern about unseen instructions. As we described in Line 400-407, unseen instructions are those instructions with (1) unseen entities and (2) unseen query semantics. However, the reviewer actually raises another definition of unseen instructions, which refers to unseen tasks such as cross instructions. We fully agree such scenarios are also a new challenge of generalization abilities, however, exploration of unseen instructions in our definition can also reveal model generalization abilities in terms of instruction semantics within each task. Therefore, we will add this discussion and provide clearer definition of unseen instructions.
>
> Responses to Questions:
>
> (A) We require both the correct number and type when evaluating in cross-instruction settings. We notice our models tend to ignore the number constraints and extract more entities in these tasks.
>
> (B) First, we use hyper-links in wiki to conduct entity linking in building our dataset, which is checked and frequently revised by contributors of wikipedia. Besides, we have considered multi-relation entity pairs in our dataset. And we train LLMs to generate a list of relations during prediction.
>
> (C) Although we split the profiling into six tasks, the evaluation metrics are calculated in kind of “entity-level” as the reviewer suggested. For example, a correct entity linking requests correctly detection of the corresponding entity mentioned first. We can also add more comprehensive entity-level overall scores in the revised version of our manuscript.

---

### Official Review · Reviewer_YbHk · 2023-08-05

**Soundness:** 3

**Excitement:**

3: Ambivalent: It has merits (e.g., it reports state-of-the-art results, the idea is nice), but there are key weaknesses (e.g., it describes incremental work), and it can significantly benefit from another round of revision. However, I won't object to accepting it if my co-reviewers champion it.

**Missing References:**

Citation for GENRE paper is wrong. Should be De Cao, Nicola, Gautier Izacard, Sebastian Riedel and Fabio Petroni. “Autoregressive Entity Retrieval.” (ICLR 2021)

**Paper Topic And Main Contributions:**

This paper proposes a new task "open-world information extraction" and constructs a dataset InstructOpenWiki for instruction-tuning models to solve this task. Although the task name seemingly includes IE as a whole, it is essentially extracting entity profiles.
They fine-tune an open-source LLM BLOOM on InstructOpenWiki and show that it outperforms non-LLM baselines and vanilla ChatGPT on the task.

**Questions For The Authors:**

A. How are the eight popular categories of instructions selected?

B. Can you clarify what is "entity priorities in Wikipedia"? Page views?

C. Is unseen ontology equivalent to an unseen entity? For example, 2023 ATP Tour was referred to as an unseen entity, but related entities such as the 2022 APT Tour could be in the training set. 2022 APT Tour would be an entity of the same type in the ontology.

**Reasons To Accept:**

- Instruction-tuning for IE is relatively understudied compared to other NLP tasks such as QA or summarization.
- The weakly-supervised dataset InstructOpenWiki could be a useful resource to help train models for entity-related tasks.
- Error analysis of different failure cases is interesting and helps break down the challenges in this task.

**Reasons To Reject:**

- The new task name is misleading and it is questionable to what degree is the task "open-domain".

The paper reads "we introduce Open-world Information Extraction (Open-world IE) to accommodate broad and diverse requests related to entity profiles surpassing predefined ontologies’ limits."  In Table 1, the example shows that the task is essentially entity linking, typing and relation extraction for the target entity. The task could simply be introduced as "entity profiling". Using the umbrella term "Open-world IE" is misleading since the task does not involve events or coreference, which are also important tasks under IE.

One of my main concerns is whether the task is truly "open-domain".  The definition of entity linking (to link a mention to an entry in a knowledge base) means that it is closed domain in nature. The only difference between GENRE and PIVOINE is that they link to different snapshots of Wikipedia. Both are closed-domain.  For entity typing and relation extraction, note that Wikidata is closed-domain and the dataset is built through linking to Wikidata. While Wikidata has a lot of entity types  (20K as reported for InstructOpenWiki), this is on the same scale as UFET [1]. Moreover, if the entity types were truly open-domain, then F1 would not be an appropriate evaluation metric as it fails to capture similarities between types such as "singer", "musical artist", and "musician".
For relations, there are only several hundred relations in the dataset. The only open-domain component seems to be the entity description.

[1] Ultra-Fine Entity Typing (Choi et al., ACL 2018)

-  The findings of the paper are not insightful.
Experiments show that fine-tuning an LLM for this instruction-guided IE task works better than non-LLM which has no instruction-following ability and ChatGPT out of the box. This conclusion seems quite obvious. (Before the LLM era, if someone said that their supervised system worked better than their zero-shot / one-shot system it would not be a meaningful finding.)
I would encourage the authors to go one step further and investigate aspects like "how much data is needed" and "why this data works". The analysis in Section 5.3  already provides some leads for investigation, for example, if failing to produce a valid JSON output is a problem, then can we solve this part directly?

**Reproducibility:**

4: Could mostly reproduce the results, but there may be some variation because of sample variance or minor variations in their interpretation of the protocol or method.

**Reviewer Confidence:**

5: Positive that my evaluation is correct. I read the paper very carefully and I am very familiar with related work.

---

> ### Author Rebuttal · Authors · 2023-08-29
>
> We sincerely appreciate the reviewer's insightful suggestions.
>
> While we initially named our task after OpenIE[1], a well-known research topic that extracts mentions and their relations in an open setting, we understand the reviewer's concern that the term IE has evolved to encompass a broader range of tasks. Therefore, we will modify our task as the entity profiling, representing the extraction of entity-related information. Similarly, we agree the open-world may be a misleading umbrella word though our initial purpose is to use it to distinguish our works with traditional closed-world IE. So we will modify the title as “Open-setting Entity Profiling” to make it more precise. Nevertheless, our work still has substantial contributions for exploring an open-set entity-related information extraction that can generalize to various semantics of instructions.
>
> We appreciate the suggestions for further exploration. Our contributions are not limited to presenting a fine-tuned task-focused LM that can outperform general proprietary chabot in zeroshot setting (which is also questionable whether ChatGPT is indeed zero-shot in our task). We explore and provide a comprehensive model, dataset, and evaluation recipe of a more open-setting information extraction task that is not well-studied in previous IE research. We consider ChatGPT as a strong baseline as there is no previous IE works that are as open as our work in entity profiling.
>
> Response to Questions:
>
> (A) We build the eight instructions based on the targeted attributions and different levels of “open-domain” in real-world requests. For example, descriptions are the most open-domain and fine-grained category while the base type can be considered as the most coarse grained request.
>
> (B) Entity priorities are calculated based on the in-link and out-link on Wikidata to reveal the popularity of entities. The detailed metric is described in Section 4.1 in [1].
>
> (C) Yes, the unseen ontology mainly refers to unseen entities in our context.
>
> [1] Andrew Chisholm and Ben Hachey. 2015. Entity Disambiguation with Web Links. Transactions of the Association for Computational Linguistics, 3:145–156.

---

### Official Review · Reviewer_FP8C · 2023-08-11

**Typos Grammar Style And Presentation Improvements:** NA
**Soundness:** 3

**Excitement:**

3: Ambivalent: It has merits (e.g., it reports state-of-the-art results, the idea is nice), but there are key weaknesses (e.g., it describes incremental work), and it can significantly benefit from another round of revision. However, I won't object to accepting it if my co-reviewers champion it.

**Missing References:**

NA

**Paper Topic And Main Contributions:**

This work considers the problem of Open-world Information Extraction (Open-world IE) and reformulates open-world IE into auto-regressive generation by linearizing the structure triples into the JSON format. Construct a substantial instruction tuning dataset for this task and finetune pre-trained BLOOM model to obtain PIVOINE. The experiments demonstrate that PIVOINE significantly outperforms previous methods, displaying impressive generalization capabilities on both unseen instructions and out-of-ontology cases.

**Questions For The Authors:**

Please refer to Reasons To Reject.

**Reasons To Accept:**

1. The paper emerges as a promising solution to tackle the open-world challenge in IE.
2. The paper constructs a dataset for Open-world IE enriched with a comprehensive corpus, extensive annotations, and diverse instructions.
3. Experimental setting is very comprehensive and insightful.
4. The paper is well-organized and clearly written.


**Reasons To Reject:**

Though the paper is well-motivated and conducts comprehensive experiments to evaluate the proposed model, I would like to provide some suggestions as follows:
1. I would like to know how to prove that the proposed method does not have data leakage issues. Maybe the INSTRUCTOPENWIKI has already seen those unknown entities, which may obviously increase the performance.
2. The method is relatively simple and mainly based on feature engineering, which may limit the interest in research community.
3. More open-source large models such as Lama, ChatGlm could also be considered.


**Reproducibility:**

3: Could reproduce the results with some difficulty. The settings of parameters are underspecified or subjectively determined; the training/evaluation data are not widely available.

**Reviewer Confidence:**

3: Pretty sure, but there's a chance I missed something. Although I have a good feel for this area in general, I did not carefully check the paper's details, e.g., the math, experimental design, or novelty.

---

> ### Author Rebuttal · Authors · 2023-08-29
>
> We sincerely appreciate the reviewer's insightful suggestions.
>
> 1. Data leakage issues are one of the main challenges in our work so we design a very delicate evaluation set to minimize the risk of data leakage. First, as we discussed in Section 4, the pretraining corpus used by BLOOM only contains Wikipedia articles before 06/20/2022 and the same of our training set, while our evaluation set only contains articles after 06/20/2022. With such a method we minimize the data leakage from the pretrain corpus and training set. Furthermore, we only evaluate our method on those new entities incorporated into Wikidata after 06/20/2022, which were rarely involved in any Wikipedia corpus before 06/20/2022. Building an evaluation set with time difference is considered as a promising way to evaluate LLMs in even more general purpose [1].
>
> 2. Our contributions are not limited to training an LLM for entity profiling but also exploring and providing a comprehensive model, dataset, and evaluation recipe of a more open-setting information extraction task that is not well-studied in previous IE research. Therefore, we believe this work would provide further impact to the community based on these substantial contributions.
>
> 3. We totally understand the reviewer's requests about evaluating more open-source LLMs on our task though LLaMa and ChatGLM are concurrent works of PIVOINE. We consider extending our method from BLOOM to more open-source LLMs as one of our valuable future works. Furthermore, we add results of the latest Vicuna-7b-v1.5 [3], which is an open-source instruction-tuning model trained on over 150k dialogues based on the foundation model LLaMa2.
> Here we present the main results of Vicuna-7b-v1.5, ChatGPT, and PIVOINE-7. Below is an extension of Table 1 in our manuscript.
>
> | Method | MD | EL(T=1) | EL(T=0.8) | ET | OpenRE | EIG(Desc.) | EIG(Alias) |
> | --- | --- | --- | --- | --- | --- | --- | --- |
> | Vicuna-7b-v1.5 | $30.6_{0.0}$ | $16.5_{0.0}$ | $17.2_{0.0}$ | $3.4_{0.0}$ | $20.0_{0.0}$ | $28.4_{0.0}$ | $10.8_{0.0}$ |
> | CHATGPT W/DEMO  | $51.1_{0.1}$ | $38.6_{0.0}$ | $39.9_{0.0}$ | $14.8_{0.0}$ | $21.4_{0.0}$ | $52.0_{0.0}$ | $13.0_{0.1}$ |
> | PIVOINE-7B  | $79.6_{0.0}$ | $69.8_{0.0}$ | $70.7_{0.0}$ | $56.4_{0.0}$ | $67.8_{0.0}$ | $80.2_{0.0}$ | $80.5_{0.0}$ |
>
> Below is an extension of Table 2 in our manuscript.
>
> | Partition | Method | MD | EL(T=1) | EL(T=0.8 ) | ET | OpenRE | EIG(Desc.) | EIG(Aliases) |
> | --- | --- | --- | --- | --- | --- | --- | --- | --- |
> |Before 05/30/2022 | Vicuna-7b-v1.5 | $26.5_{0.0}$ | $14.9_{0.0}$ | $15.1_{0.0}$ | $3.3_{0.0}$ | $36.4_{0.0}$ | $29.4_{0.0}$ | $8.8_{0.0}$ |
> |After 05/30/2022 | Vicuna-7b-v1.5 | $26.7_{0.0}$ | $13.0_{0.0}$ | $14.3_{0.0}$ | $3.1_{0.0}$ | $25.5_{0.0}$ | $25.8_{0.0}$ | $10.5_{0.0}$ |
>
> Results of Vicuna-7b-v1.5 show PIVOINE can outperform existing powerful open-source instruction-tuning LLMs, which provides further evidence of effectiveness of our method. We will add more comprehensive comparison experiments, such as involving Alpaca, in the revised version of the manuscript.
>
>
>
> [1] Jang, Joel, et al. "TemporalWiki: A lifelong benchmark for training and evaluating ever-evolving language models." arXiv preprint arXiv:2204.14211 (2022).

---

### Official Review · Reviewer_CAmu · 2023-08-12

**Soundness:** 2

**Excitement:**

3: Ambivalent: It has merits (e.g., it reports state-of-the-art results, the idea is nice), but there are key weaknesses (e.g., it describes incremental work), and it can significantly benefit from another round of revision. However, I won't object to accepting it if my co-reviewers champion it.

**Paper Topic And Main Contributions:**

This paper tackles the new problem of entity-centric Information Extraction, which goes beyond the conventional open IE tasks by considering user instructions. To handle this task, this paper creates a substantial instruction tuning dataset and s fine-tunes a large language model named PIVOINE on this dataset. The results demonstrate PIVOINE's superiority over traditional methods and ChatGPT-based baselines in handling unseen instructions and out-of-ontology cases.

**Reasons To Accept:**

(1) Good Writing: The paper is well-written, presenting the concepts and the ideas in a clear and concise manner, making it accessible to a broad readership.

(2) New Problem: The authors have targeted an relatively unexplored area, considering user instructions for open IE. This novel focus has the potential to offers new opportunities for research.

(3) New Dataset: The constructed dataset is a substantial contribution, providing a comprehensive corpus and diverse instructions specifically tailored for the proposed task. This dataset can serve as a valuable resource for future research.


**Reasons To Reject:**

(1) Overclaimed task setting: this paper claims the task as open world information extraction, but actually it only focuses on the entity-related tasks. It cannot even handle some existing well-defined ie tasks (like slot filling, coreference resolution, and table extraction), let alone solve more open-scenario ie problems. Thus, it’s not convincing to claim the proposed task as open-world.

Additionally, this paper claims that the novelty of open-world IE is considering user instructions and going beyond a predefined ontology of entities and relations. However,  many existing work on open information extraction can already achieve this goal. Thus,  this point is kind of overclaimed.

(2) Data leakage problem: Since the dataset is collected from Wikipedia, there might be a risk of data leakage if the large language models have already been trained on the test data. This could lead to overly optimistic performance metrics and hinder the generalization of the model to truly unseen data.

(3) Missing comparing experiment: this paper targets on instruction tuning for IE. There are existing powerful open-source instruction tuning models like Alpaca [1]. Without comparing with these models, it would not be persuasive enough about the effectiveness of the proposed method.


**Reproducibility:**

4: Could mostly reproduce the results, but there may be some variation because of sample variance or minor variations in their interpretation of the protocol or method.

**Reviewer Confidence:**

5: Positive that my evaluation is correct. I read the paper very carefully and I am very familiar with related work.

---

> ### Author Rebuttal · Authors · 2023-08-29
>
> We thank the reviewer for the valuable feedback.
> While we initially named our task after OpenIE[1], a well-known research topic that extracts mentions and their relations in an open setting, we understand the reviewer's concern that the term IE has evolved to encompass a broader range of tasks. Therefore, we will modify our task as the entity profiling, representing the extraction of entity-related information. Similarly, we agree the open-world may be a misleading umbrella word though our initial purpose is to use it to distinguish our works with traditional closed-world IE. So we will modify the title as “Open-setting Entity Profiling” to make it more precise. Nevertheless, our work still has substantial contributions for exploring an open-set entity-related information extraction that can generalize to various semantics of instructions.
>
> And we also disagree with the reviewer’s claim that many existing works on open information extraction can achieve this goal. Previous works in OpenIE cannot handle instructions as varied as in our work. Recent works such as InstructUIE[2] also try to extend OpenIE to  handle user requests in multiple IE tasks but they are concurrent works of PIVOINE. We would sincerely appreciate it if the reviewer can provide further related works for reference.
> We fully understand the reviewer’s concern about data leakage in LLM pretraining, so we have provided analysis on this issue in our manuscript. We consider the potential leakage may come from corpus or entities. First, we show the pretraining corpus used in BLOOM is before 06/20/2022 in the footnote in Page 5, while our evaluation set only contains corpus after 06/20/2022, which can be considered independent with the pretraining corpus. Second, we only consider extracting new entities after 06/20/2022 to minimize the risk of data leakage.
> We agree comparing experiments with aligned LLMs such as Alpaca will further elaborate the effectiveness of our methods. We add results of the latest Vicuna-7b-v1.5 [3], which is an open-source instruction-tuning model trained on over 150k dialogues based on the foundation model LLaMa2.
> Here we present the main results of Vicuna-7b-v1.5, ChatGPT, and PIVOINE-7. Below is an extension of Table 1 in our manuscript.
>
> | Method | MD | EL(T=1) | EL(T=0.8) | ET | OpenRE | EIG(Desc.) | EIG(Alias) |
> | --- | --- | --- | --- | --- | --- | --- | --- |
> | Vicuna-7b-v1.5 | $30.6_{0.0}$ | $16.5_{0.0}$ | $17.2_{0.0}$ | $3.4_{0.0}$ | $20.0_{0.0}$ | $28.4_{0.0}$ | $10.8_{0.0}$ |
> | CHATGPT W/DEMO  | $51.1_{0.1}$ | $38.6_{0.0}$ | $39.9_{0.0}$ | $14.8_{0.0}$ | $21.4_{0.0}$ | $52.0_{0.0}$ | $13.0_{0.1}$ |
> | PIVOINE-7B  | $79.6_{0.0}$ | $69.8_{0.0}$ | $70.7_{0.0}$ | $56.4_{0.0}$ | $67.8_{0.0}$ | $80.2_{0.0}$ | $80.5_{0.0}$ |
>
> Below is an extension of Table 2 in our manuscript.
>
> | Partition | Method | MD | EL(T=1) | EL(T=0.8 ) | ET | OpenRE | EIG(Desc.) | EIG(Aliases) |
> | --- | --- | --- | --- | --- | --- | --- | --- | --- |
> |Before 05/30/2022 | Vicuna-7b-v1.5 | $26.5_{0.0}$ | $14.9_{0.0}$ | $15.1_{0.0}$ | $3.3_{0.0}$ | $36.4_{0.0}$ | $29.4_{0.0}$ | $8.8_{0.0}$ |
> |After 05/30/2022 | Vicuna-7b-v1.5 | $26.7_{0.0}$ | $13.0_{0.0}$ | $14.3_{0.0}$ | $3.1_{0.0}$ | $25.5_{0.0}$ | $25.8_{0.0}$ | $10.5_{0.0}$ |
>
> Results of Vicuna-7b-v1.5 show PIVOINE can outperform existing powerful open-source instruction-tuning LLMs, which provides further evidence of effectiveness of our method. We will add more comprehensive comparison experiments, such as involving Alpaca, in the revised version of the manuscript.
>
>
> [1] Niklaus, Christina, et al. "A survey on open information extraction." arXiv preprint arXiv:1806.05599 (2018).
> [2] Wang, Xiao, et al. "InstructUIE: Multi-task Instruction Tuning for Unified Information Extraction." arXiv preprint arXiv:2304.08085 (2023).
> [3] Zheng, Lianmin, et al. "Judging LLM-as-a-judge with MT-Bench and Chatbot Arena." arXiv preprint arXiv:2306.05685 (2023).

---

### Meta-Review · Area_Chair_F2Qf · 2023-09-19

**Recommendation:** 3

**Metareview:**

This paper presents INSTRUCTOPENWIKI, a large-scale instruction-following open-world IE dataset, and the creation of the PIVOINE model, an instruction tuning model based on BLOOM. The reformulation of open-world IE into a linearized JSON format is a promising solution to the task, though the open-world naming is overclaimed and misleading. The main concerns are with data leakage, though this issue seem to be addressed by the authors in the rebuttal.

---

### Decision · Program_Chairs · 2023-10-07

**Decision:**

Accept-Findings

**Comment:**

This paper presents INSTRUCTOPENWIKI, a large-scale instruction-following open-world IE dataset, and the creation of the PIVOINE model, an instruction tuning model based on BLOOM. The reformulation of open-world IE into a linearized JSON format is a promising solution to the task, though the open-world naming is overclaimed and misleading. The main concerns are with data leakage, though this issue seem to be addressed by the authors in the rebuttal.